# Context Mover's Distance & Barycenters: Optimal transport of contexts for building representations

**Sidak Pal Singh, Andreas Hug, Aymeric Dieuleveut & Martin Jaggi**
EPFL
Switzerland
`{sidak.singh,andreas.hug,aymeric.dieuleveut,martin.jaggi}@epfl.ch`

## Abstract

We present a framework for building unsupervised representations of entities and their compositions, where each entity is viewed as a probability distribution rather than a fixed length vector. In particular, this distribution is supported over the contexts which co-occur with the entity and are embedded in a suitable low-dimensional space. This enables us to consider the problem of representation learning with a perspective from Optimal Transport and take advantage of its numerous tools such as Wasserstein distance and Wasserstein barycenters. We elaborate how the method can be applied for obtaining unsupervised representations of text and illustrate the performance quantitatively as well as qualitatively on tasks such as measuring sentence similarity and word entailment, where we empirically observe significant gains (e.g., 4.1% relative improvement over Sent2vec, GenSen).

The key benefits of the proposed approach include: (a) capturing uncertainty and polysemy via modeling the entities as distributions, (b) utilizing the underlying geometry of the particular task (with the ground cost), (c) simultaneously providing interpretability with the notion of optimal transport between contexts and (d) easy applicability on top of existing point embedding methods. In essence, the framework can be useful for any unsupervised or supervised problem (on text or other modalities); and only requires a co-occurrence structure inherent to many problems. The code, as well as pre-built histograms, are available under https://github.com/context-mover/.

## 1 Introduction

One of the driving factors behind recent successes in machine learning has been the development of better methods for data representation, thus forming the foundation around which rest of the model architecture gets built. Examples include continuous vector representations for language (Mikolov et al., 2013; Pennington et al., 2014), convolutional neural network based feature representations for images and text (LeCun et al., 1998; Collobert & Weston, 2008; Kalchbrenner et al., 2014), or via the hidden state representations of LSTMs (Hochreiter & Schmidhuber, 1997; Sutskever et al., 2014). Pre-trained unsupervised representations in particular have been immensely useful as general purpose features for model initialization (Kim, 2014), downstream tasks, (Severyn & Moschitti, 2015; Deriu et al., 2017) and in domains with limited supervised information (Qi et al., 2018).

The shared idea across these methods is to map input entities to dense vector embeddings lying in a low-dimensional latent space where the semantics of inputs are preserved. Thus, each entity of interest (e.g., a word) is represented directly as a single point (i.e., its embedding vector) in space, which is typically Euclidean.

In contrast, we approach the problem of building unsupervised representations in a fundamentally different manner. We focus on the co-occurrence information between the entities and their contexts, and represent each entity as a *probability distribution (histogram) over its contexts*. Here the contexts themselves are embedded as points in a suitable low-dimensional space. This allows us to cast finding distance between entities as an instance of the *Optimal Transport problem* (Monge, 1781;

Kantorovich, 1942; Villani, 2008). So, our resulting framework intuitively compares the cost of moving the contexts of a given entity to the contexts of another, which motivates the naming *Context Mover's Distance* (CMD).

We will call this distribution over contexts embeddings the *distributional estimate* of our entity of interest (see Figure 1), while we refer to the individual embeddings of contexts as *point estimates*. More precisely, the contexts refer to any generic entities or objects (such as words, phrases, sentences, images, etc.) co-occurring with the entities to be represented.

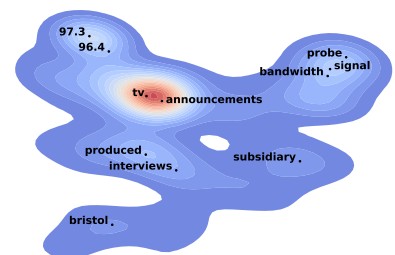

The main motivation for our proposed approach originates from the domain of natural language, where the entities (words, phrases, or sentences) generally have different semantics depending on the context under which they are present. Hence, it is important to consider representations that are able to effectively capture such inherent uncertainty and polysemy, and

Figure 1: Distributional estimate for the entity 'radio'.

we will argue that distributional estimates capture more of this information compared to point-wise embedding vectors alone. In particular, we will see that the co-occurrence information required to build the distributions is already obtained as the first step of point-wise embedding methods, like in GloVe (Pennington et al., 2014), but has largely been ignored in the past.

Further, this co-occurrence information that is the crucial building block of our approach is inherent to a wide variety of problems, for instance, recommending products such as movies or web-advertisements (Grbovic et al., 2015), nodes in a graph (Grover & Leskovec, 2016), sequence data, or other entities (Wu et al., 2017). This means that, in principle, our framework can be employed to obtain a representation of various entities present across these problems.

Overall, we strongly advocate for representing entities with distributional estimates due to the above stated reasons. But at the same time, our message isn't that point-wise embedding methods should cease to exist, rather that both kinds of methods should go hand in hand. This will be reflected through building distributional estimates on the top of existing point embedding methods, as well as how we can combine them (cf. Section 4) to get the best of these intrinsically different ideas.

Lastly, the connection to optimal transport at the level of entities and contexts paves the way to make better use of its vast toolkit (like Wasserstein distances, barycenters, barycentric coordinates, etc.) for applications in NLP, which in the past has primarily been restricted to document distances of original words (Kusner et al., 2015; Huang et al., 2016), as opposed to contexts. Thanks to the entropic regularization introduced by Cuturi (2013), optimal transport computations can be carried out efficiently in a parallel and batched manner on GPUs.

**Contributions:**   1) Employing the notion of optimal transport of contexts as a distance measure, we illustrate how our framework can be of benefit for various important tasks, including word and sentence representations, sentence similarity, as well as hypernymy (entailment) detection. The method is static and does not require any additional learning, and can be readily used on top of existing embedding methods.

2) The resulting representations, as portrayed in Figures 1, 3, 4, capture the various senses under which the entity occurs. Next, the transport map obtained through CMD (see Figure 2) gives a clear interpretation of the resulting distance obtained between two entities.

3) Our Context Mover's Distance (CMD) can be used to measure any kind of distance (even asymmetric) between words, by defining a suitable underlying cost on the movement of contexts, which we show can lead to a state-of-the-art metric for word entailment.

4) Defining the transport over contexts has the additional benefit that the representations are compositional - they directly extend from entities to groups of entities (of any size), such as from word to sentence representations. To this end, we utilize the notion of Wasserstein barycenters, which to the best of our knowledge has never been considered in the past. This results in a significant performance boost on multiple datasets, and even outperforming supervised methods like InferSent (Conneau et al., 2017) and GenSen (Subramanian et al., 2018) by a decent margin.

## 2 RELATED WORK

**Vector representations.** The idea of using vector space models for natural language dates back to Bengio et al. (2003), but in particular has been popularized by the seminal works of Word2vec (Mikolov et al., 2013) and GloVe (Pennington et al., 2014). Further, works such as (Levy & Goldberg, 2014a; Bojanowski et al., 2016) have suggested to enrich these embeddings to capture additional information. One of the problems that still persists is the inability to capture, within just a point embedding, the various semantics and uncertainties associated with the occurrence of a particular word (Huang et al., 2012; Guo et al., 2014).

**Representing with distributions.** This line of work is fairly recent, mainly originating from Vilnis & McCallum (2014), who proposed to represent words with Gaussian distributions, and later extended to mixtures of Gaussians in (Athiwaratkun & Wilson, 2017). Concurrent to this work, Muzellec & Cuturi (2018) and Sun et al. (2018) have suggested using elliptical and Gaussian distributions endowed with a Wasserstein metric respectively. While these methods already provide richer information than typical vector embeddings, their form restricts what could be gained by allowing for arbitrary distributions as possible here. Our proposal of distributional estimate (i.e., distribution over context embeddings), inherently relies upon the empirically obtained co-occurrence information of a word and its contexts. Hence, this naturally allows for the use of optimal transport (or Wasserstein metric) in the space containing the contexts, and leads to an interpretation (Figure 2) which is not available in the above approaches. Another consequence is that the training procedure required in these methods is not necessary for our approach, as we can just utilize the existing point-embedding methods together with the co-occurrence information.

Apart from embedding entities in the Wasserstein space, other metric spaces like the Hyperbolic space, have recently gained attention for modelling hierarchical structures (Nickel & Kiela, 2017; Ganea et al., 2018). But, these are so far restricted to supervised tasks[1], not allowing unsupervised representation learning, which is the focus here.

**Optimal Transport in NLP.** The primary focus of the explorations of optimal transport in NLP has been on transporting words directly, and for downstream applications rather than representation learning in general. These include document distances (Kusner et al., 2015; Huang et al., 2016), topic modelling (Rolet et al., 2016; Xu et al., 2018), document clustering (Ye et al., 2017), and others (Zhang et al., 2017; Grave et al., 2018). For example, the Word Mover's Distance (WMD; Kusner et al., 2015) considers computing the distance between documents as an optimal transport between their bag-of-words, and in itself doesn't lead to a representation. When the transport is defined at the level of words like in these approaches, it can not be used to represent words themselves. In our approach, the transport is considered over contexts instead, which enables us to develop representations for words and also extend them to represent composition of words (i.e., sentences, documents) in a principled manner, as will be illustrated further through the examples of entailment detection and sentence representation respectively.

## 3 BACKGROUND ON OPTIMAL TRANSPORT

Optimal Transport (OT) provides a way to compare two probability distributions defined over a space $\mathcal{G}$ (commonly known as the ground space), given an underlying distance or more generally a cost of moving one point to another in the ground space. In other terms, it lifts a distance between points to a distance between distributions. Other divergences, such as Kullback-Leibler (KL), or $f$-divergence in general, only focus on the probability mass values, thus ignoring the geometry of the ground space: something which we utilize throughout this work via OT. Also, $\text{KL}(\mu||\nu)$ is defined only when the distribution $\mu$ is absolutely continuous with respect to $\nu$. Having motivated our choice, we give a short yet formal background on OT in the discrete case.

**Linear Program Formulation.** Consider an empirical probability measure of the form $\mu = \sum_{i=1}^{n} a_i \delta(\boldsymbol{x}^{(i)})$ where $X = (\boldsymbol{x}^{(1)}, \ldots, \boldsymbol{x}^{(n)}) \in \mathcal{G}^n$, $\delta(\boldsymbol{x})$ denotes the Dirac (unit mass) distribution at point $\boldsymbol{x} \in \mathcal{G}$, and $(a_1, \ldots, a_n)$ lives in the probability simplex $\Sigma_n := \left\{ \boldsymbol{p} \in \mathbb{R}_+^n \mid \sum_{i=1}^{n} p_i = 1 \right\}$. Now given a second empirical measure, $\nu = \sum_{j=1}^{m} b_j \delta(\boldsymbol{y}^{(j)})$, with $Y = (\boldsymbol{y}^{(1)}, \ldots, \boldsymbol{y}^{(m)}) \in \mathcal{G}^m$,

---

[1]Also, similar for the elliptical embeddings (Muzellec & Cuturi, 2018) in the case of Hypernymy.

and $(b_1, \ldots, b_m) \in \Sigma_m$, and if the ground cost of moving from point $\boldsymbol{x}^{(i)}$ to $\boldsymbol{y}^{(j)}$ is denoted by $M_{ij}$, then the Optimal Transport distance between $\mu$ and $\nu$ is the solution to the following linear program.

$$\mathrm{OT}(\mu, \nu; M) := \min_{\substack{T \in \mathbb{R}_+^{n \times m} \\ \text{s.t. } \forall i, \sum_j T_{ij} = a_i, \ \forall j, \sum_i T_{ij} = b_j}} \sum_{ij} T_{ij} M_{ij} \tag{1}$$

Here, the optimal $T \in \mathbb{R}_+^{n \times m}$ is referred to as the *transportation matrix*: $T_{ij}$ denotes the optimal amount of mass to move from point $\boldsymbol{x}^{(i)}$ to point $\boldsymbol{y}^{(j)}$. Intuitively, OT is concerned with the problem of moving a given supply of goods from certain factories to meet the demands at some shops, such that the overall transportation cost is minimal.

**Distance.** When $\mathcal{G} = \mathbb{R}^d$ and the cost is defined with respect to a metric $D_{\mathcal{G}}$ over $\mathcal{G}$ (i.e., $M_{ij} = D_{\mathcal{G}}(\boldsymbol{x}^{(i)}, \boldsymbol{y}^{(j)})^p$ for any $i, j$), OT defines a distance between empirical probability distributions. This is the $p$-Wasserstein distance, defined as $\mathcal{W}_p(\mu, \nu) := \mathrm{OT}(\mu, \nu; D_{\mathcal{G}}^p)^{1/p}$. In most cases, we are only concerned with the case where $p = 1$ or $2$.

**Regularization and Sinkhorn iterations.** The cost of exactly solving OT scales at least in $\mathcal{O}(n^3 \log(n))$ ($n$ being the cardinality of the support of the empirical measure) when using network simplex or interior point methods. Following Cuturi (2013), we consider the entropy regularized Wasserstein distance, $\mathcal{W}_{p,\lambda}(\mu, \nu) := \mathrm{OT}_\lambda(\mu, \nu; D_{\mathcal{G}}^p)^{1/p}$, where the search space for the optimal $T$ is instead restricted to a smooth solution close to the extreme points of this linear program, as follows:

$$\mathrm{OT}_\lambda(\mu, \nu; M) := \min_{T \in \mathbb{R}_+^{n \times m}} \sum_{ij} T_{ij} M_{ij} - \lambda H(T), \tag{2}$$
$$\text{s.t. } \begin{cases} \forall i, \sum_j T_{ij} = a_i, \\ \forall j, \sum_i T_{ij} = b_j \end{cases}$$

where $H(T) = -\sum_{ij} T_{ij} \log T_{ij}$. The regularized problem can then be solved efficiently using Sinkhorn iterations (Sinkhorn, 1964), albeit at the cost of some approximation error. This can be controlled by the regularization strength $\lambda \geq 0$, with the true OT recovered at $\lambda = 0$. While the cost of each Sinkhorn iteration is quadratic in $n$, Altschuler et al. (2017) have shown that the convergence to an $\epsilon$-accurate solution can be attained in a number of iterations that is independent of $n$, thus resulting in an overall complexity of $\widetilde{O}(n^2/\epsilon^3)$.

**Barycenters.** In Section 6, we will make use of the notion of averaging in the Wasserstein space. More precisely, the Wasserstein barycenter, introduced by Agueh & Carlier (2011), is a probability measure that minimizes the sum of ($p$-th power) Wasserstein distances to the given measures. Formally, given $N$ measures $\{\nu_1, \ldots, \nu_N\}$ with corresponding weights $\boldsymbol{\eta} = \{\eta_1, \ldots, \eta_N\} \in \Sigma_N$, the Wasserstein barycenter can be written as

$$\mathcal{B}_p(\nu_1, \ldots, \nu_N) = \arg\min_\mu \sum_{i=1}^N \eta_i \mathcal{W}_p(\mu, \nu_i)^p. \tag{3}$$

For practical purposes, we consider the regularized barycenter $\mathcal{B}_{p,\lambda}$, using entropy regularized Wasserstein distances $\mathcal{W}_{p,\lambda}$ in the above minimization problem, following Cuturi & Doucet (2014). Employing the iterative Bregman projections (Benamou et al., 2015), we obtain an approximation of the solution at a reasonable computational cost.

## 4 METHODOLOGY

In this section, we define the distributional estimate that we use to represent each entity. Since we take the guiding example of building text representations, we consider each entity to be a word for simplicity.

**Distributional Estimate** ($\mathbb{P}_V^w$). For a word $w$, its distributional estimate is built from a histogram $\mathrm{H}^w$ over the set of contexts $\mathcal{C}$, and an embedding of these contexts into a space $\mathcal{G}$. The histogram essentially measures how likely it is for a word $w$ to occur in a particular context $c$, i.e., probability $p(w|c)$. The exact formulation of this in closed form is generally intractable and hence it's common to empirically estimate this by the number of occurrences of the word $w$ in context $c$, relative to the total frequency of context $c$ in the corpus.

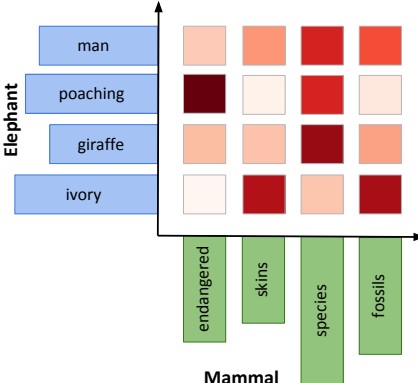

Figure 2: *Illustration of Context Mover's Distance (CMD) (Eq. (5)) between elephant & mammal (when represented with their distributional estimates and using entailment ground metric discussed in Section 7). Here, we pick four contexts at random from their top 20 contexts in terms of PPMI. The square cells above denote the entries of the transportation matrix (or transport map) $T$ obtained in the process of computing CMD. The darker a cell, the larger the amount of mass moved between the corresponding contexts.*

Thus one natural way to build this histogram is to maintain a co-occurrence matrix between words in our vocabulary and all possible contexts, where each entry indicates how often a word and context occur in a window of fixed size $L$. Then, the bin values $(\mathrm{H}^w)_{c \in \mathcal{C}}$ of the histogram can be viewed as the row corresponding to $w$ in this co-occurrence matrix.

Next, the simplest embedding of contexts is into the space of one-hot vectors of all the possible contexts. However, this induces a lot of sparsity in the representation and the distance between such embeddings of contexts does not reflect their semantics. A classical solution would be to instead find a dense low-dimensional embedding of contexts that captures the semantics, possibly using techniques such as SVD or deep neural networks. We denote by $V = (\mathbf{v}_c)_{c \in \mathcal{C}}$ an embedding of the contexts into this low-dimensional space $\mathcal{G} \subset \mathbb{R}^d$, which we refer to as the *ground space*

Combining the histogram $\mathrm{H}^w$ and the context embeddings $V$, we represent the word $w$ by the following empirical distribution, referred to as the *distributional estimate* of the word:

$$\mathbb{P}_V^w := \sum_{c \in \mathcal{C}} (\mathrm{H}^w)_c \, \delta(\mathbf{v_c}). \tag{4}$$

**Distance.** If we equip the ground space $\mathcal{G}$ with a meaningful metric $D_{\mathcal{G}}$ and use distributional estimates $(\mathbb{P}_V^w)$ to represent the words, then we can define a distance between two words $w_i$ and $w_j$ as the solution to the following optimal transport problem:

$$\mathrm{CMD}(w_i, w_j; D_{\mathcal{G}}^p) := \mathrm{OT}(\mathbb{P}_V^{w_i}, \mathbb{P}_V^{w_j}; D_{\mathcal{G}}^p) \simeq \mathcal{W}_{p,\lambda}(\mathbb{P}_V^{w_i}, \mathbb{P}_V^{w_j})^p. \tag{5}$$

We call this distance in Eq.(5) as the *Context Mover's Distance* (CMD).

**Intuition.** Two words are similar in meaning if the contexts of one word can be easily transported to the contexts of the other, with this cost of transportation being measured by $D_{\mathcal{G}}$. This idea still remains in line with the distributional hypothesis (Harris, 1954; Rubenstein & Goodenough, 1965) that words in similar contexts have similar meanings, but provides a precise way to quantify it.

**Interpretation.** The particular definition of CMD in Eq.(5), lends a pleasing interpretation (see Figure 2 in terms of the transportation map $T$. This interpretation can be useful in understanding why and how are the two words being considered as similar in meaning, by looking at this movement of contexts.

**Relating neural and count-based models.** While the histogram information is characteristic of count-based language models, the point estimates are reflective of embeddings in neural language models. But in the distributional estimate which underlies CMD, both these elements are closely tied together. Since $\mathrm{CMD}(w_t, w_{1...t-1})$ [2] can be interpreted as giving unnormalized negative log probabilities for a language model, we see how this combines neural and count-based models.

**Mixed Distributional Estimate.** We also consider adding the information from point estimate into the distributional estimate to get best of both the worlds. This is done by adding a point estimate (i.e., a Dirac at its location) as an additional context with a particular mixing weight, denoted as $m$. The other contexts in the distributional estimate are reweighted to sum to $1 - m$.

---

[2]With past words in the sequence, $w_1, \ldots, w_{t-1}$, aggregated via the barycenter, Eq.(3, 6).

**Roadmap.** Next, we discuss concretely how this framework can be applied and for brevity we restrict to the particular case where contexts consist of single words. Section 6 details how this framework can be extended to obtain a representation for the composition of entities via Wasserstein barycenter. Lastly in section 7, we utilize the fact that the CMD in Eq.(5) is parameterized by ground cost, and show how this flexibility can be used to define an asymmetric cost measuring entailment.

## 5 CONCRETE FRAMEWORK

**Making associations better.** We consider that a word and a context word co-occur if the latter appears in a symmetric window of size $L$ around the word whose distributional estimate we seek (i.e., the target word). But, it is commonly understood that co-occurrence counts alone may not necessarily suggest a strong association between the two. The well-known Positive Pointwise Mutual Information (PPMI) matrix (Church & Hanks, 1990; Levy et al., 2015) addresses this shortcoming, and is defined as follows: $\text{PPMI}(w, c) := \max(\log(\frac{p(w,c)}{p(w) \times p(c)}), 0)$. This means that the PPMI entries are non-zero when the joint probability of target and context words co-occurring is higher than the probability when they are independent. Typically, these probabilities are estimated from the co-occurrence counts in the corpus. Further improvements to the PPMI matrix have been suggested, like in Levy & Goldberg (2014b), and following them we make use of a shifted and smoothed PPMI matrix, denoted by $\text{SPPMI}_{\alpha,s}$ where $\alpha$ and $s$ denote the smoothing and k-shift parameters[3]. Overall, these variants of PPMI enable us to extract better semantic associations from the co-occurrence matrix. Hence, the bin values (at context $c$) for the histogram of word $w$ in Eq. (4) can be formulated as: $(\mathbf{H}^w)_c := \frac{\text{SPPMI}_{\alpha,s}(w,c)}{\sum_{c \in \mathcal{C}} \text{SPPMI}_{\alpha,s}(w,c)}$.

**Computational considerations.** A natural question could arise that CMD might be computationally intractable in its current formulation, as the possible number of contexts can be enormous. Since the contexts are mapped to dense embeddings, it is possible to only consider $K$ *representative contexts* (centroids of the clusters of contexts), each covering some part $\mathcal{C}_k$ of the set of contexts $\mathcal{C}$. The histogram for word $w$ with respect to these contexts can then be written as $\tilde{\mathbb{P}}^w_{\tilde{V}} = \sum_{k=1}^K (\tilde{\mathbf{H}}^w)_k \, \delta(\tilde{\mathbf{v}}_k)$, where $\tilde{\mathbf{v}}_k \in \tilde{V}$ is the point estimate of the $k^{th}$ representative context, and $(\tilde{\mathbf{H}}^w)_k$ denotes the new histogram bin values (formed by combining the SPPMI contributions). Precise definitions and a detailed discussion on the effect of the number of clusters are given in the supplementary Section S2.

**Overall efficiency.** With the above aspects in account and using batched implementations on (Nvidia TitanX) GPUs, it is possible to compute around **13,700** Wasserstein-distances/second (for histogram of size 100). Same also holds for barycenters, where we can compute **4,600** barycenters/second for sentences of length 25 and histogram size of 100. Building the histogram information comes almost for free during the typical learning of embeddings, as in GloVe (Pennington et al., 2014).

## 6 SENTENCE REPRESENTATIONS

The goal of this task is to develop a representation for sentences, that captures the semantics conveyed by it. Most unsupervised representations proposed in the past rely on the composition of vector embeddings for the words, through either additive, multiplicative, or other ways (Mitchell & Lapata, 2008; Arora et al., 2017; Pagliardini et al., 2017). As before, our aim is to represent sentences as distributional estimates to better capture the inherent uncertainty and polysemy.

We hypothesize that a sentence, $S = (w_1, w_2, \dots, w_N)$, can be efficiently represented via the Wasserstein barycenter of the distributional estimates of its words,

$$\tilde{\mathbb{P}}_S := \mathcal{B}_{p,\lambda} \left( \tilde{\mathbb{P}}^{w_1}_V, \tilde{\mathbb{P}}^{w_2}_V, \dots, \tilde{\mathbb{P}}^{w_N}_V \right). \tag{6}$$

The motivation is that since the barycenter minimizes the sum of optimal transports, cf. Eq. (3), it should result in a representation which best captures the simultaneous occurrence of the words in a sentence. For instance, consider two probability measures which are Diracs, $\delta(\boldsymbol{x})$ and $\delta(\boldsymbol{y})$, with equal weights and under Euclidean ground metric. Then, the Wasserstein barycenter is $\delta(\frac{\boldsymbol{x}+\boldsymbol{y}}{2})$ while simple averaging gives $\frac{1}{2}(\delta(\boldsymbol{x}) + \delta(\boldsymbol{y}))$. In fact, Figure 3 says it all, and compares these two kinds of averaging based on the actual distributional estimates of the words. Hence, illustrating

---

[3]Please refer to Appendix S1.2 for the definition of SPPMI and more details such as our column normalization.

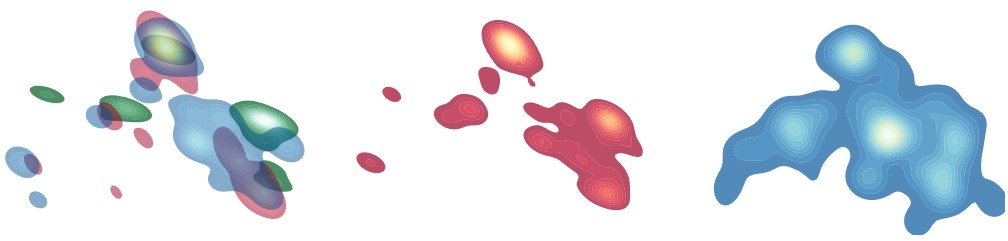

(a) Distributional estimates of *'my'* (green), *'love'* (red) and *'life'* (blue).

(b) Euclidean average.

(c) Wasserstein barycenter.

Figure 3: Illustrates how Wasserstein barycenter takes into account the geometry of ground space, while the Euclidean averaging just focuses on the probability mass.[5]

that Wasserstein barycenter is better suited than naive averaging, for applications having an innate geometry. We refer to this representation as the *Context Mover's Barycenters* (CoMB) henceforth.

Averaging of point-estimates, in many variants (Iyyer et al., 2015; Arora et al., 2017; Pagliardini et al., 2017), has been shown to be surprisingly effective for multiple NLP tasks including sentence similarity. Interestingly, this can be seen as a special case of CoMB, when the distribution associated to a word is just a Dirac at its point estimate. It becomes apparent that having a rich distributional estimate for a word could turn out to be advantageous.

Since with CoMB (Eq. (6)), each sentence is also a distribution over the ground space $\mathcal{G}$ containing the contexts, we can utilize the Context Mover's Distance (CMD) defined in Eq. (5) to define the distance between two sentences $S_1$ and $S_2$, under a given ground metric $D_{\mathcal{G}}$ as follows,

$$\mathrm{CMD}(S_1, S_2; D_{\mathcal{G}}^p) := \mathrm{OT}(\tilde{\mathbb{P}}_V^{S_1}, \tilde{\mathbb{P}}_V^{S_2}; D_{\mathcal{G}}^p) \simeq \mathcal{W}_{p,\lambda}(\tilde{\mathbb{P}}_V^{S_1}, \tilde{\mathbb{P}}_V^{S_2})^p. \quad (7)$$

## 6.1 EXPERIMENTAL SETUP

To evaluate CoMB as an effective sentence representation, we consider 24 datasets from SemEval semantic textual similarity (STS) tasks (Agirre et al., 2012; 2013; 2014; 2015; 2016), containing sentences from domains such as news headlines, forums, Twitter, etc. The objective here is to give a similarity score to each sentence pair and rank them, which is evaluated against the ground truth ranking via Pearson correlation.

As a ground metric ($D_{\mathcal{G}}$), we consider Euclidean or angular distance between the point estimates (depending upon validation performance). The point estimates are obtained by using GloVe (Pennington et al., 2014) on the Book Corpus (Zhu et al., 2015), and via this we also get the histogram information needed for the distributional estimate. The representative contexts are obtained by performing K-means clustering of the point estimates with respect to angular distance.

We benchmark[6] against a variety of unsupervised methods such as Neural Bag-of-Words (NBoW) averaging of point estimates, SIF from Arora et al. (2017) who regard it as a "simple but tough-to-beat baseline" and utilize weighted NBoW averaging with principal component removal, Sent2vec (Pagliardini et al., 2017), Skip-thought (Kiros et al., 2015), and Word Mover's Embedding (WME; Wu et al., 2018) which is a recent variant of WMD. For comparison, we also show the performance of recent supervised methods such as InferSent (Conneau et al., 2017) and GenSen (Subramanian et al., 2018), although these methods are clearly at an advantage due to training on labeled corpora.

## 6.2 EMPIRICAL RESULTS

**Ground Metric: GloVe.** Table 1 (a) shows the performance of CoMB when GloVe embeddings trained on Book Corpus are used as ground metric. Hyperparameters for all the methods are tuned on STS16, and the best configuration so obtained is used for the other STS tasks. We observe that

---

[5]For visualization purposes in Figures 1, 3, 4, we compute a two dimensional representation of the actual context emb using t-SNE (Maaten & Hinton, 2008) and use a kernel density estimate to smooth the distributions.

[6]We use SIF's publicly available implementation (https://github.com/PrincetonML/SIF) and adapted version of SentEval (Conneau & Kiela, 2018) for evaluating CoMB.

| Model | Corpus | Val. Set STS16 | Test Set STS12 | STS13 | STS14 | STS15 | Avg. |
|---|---|---|---|---|---|---|---|
| *(a) Unsupervised methods with **GloVe** embeddings* | | | | | | | |
| NBoW | | 19.2 | 21.1 | 13.5 | 25.0 | 30.7 | 22.6 |
| SIF | | 26.6 | 32.4 | 23.0 | 34.1 | 35.3 | 31.2 |
| SIF $_{PC\ removed}$ | Book Corpus | 57.6 | 41.0 | 50.1 | 51.9 | 52.8 | 49.0 |
| CoMB | | 52.4 | 48.2 | 42.2 | 54.9 | 53.8 | 49.8 |
| CoMB $_{Mix}$ | | 60.2 | 50.5 | 51.0 | 58.3 | 60.5 | 55.1 |
| CoMB $_{Mix + PC\ removed}$ | | 63.0 | 49.3 | 56.5 | 60.8 | 64.0 | 57.7 |
| *(b) Unsupervised methods with **Sent2vec** embeddings* | | | | | | | |
| Sent2vec | | 69.1 | 55.6 | 57.1 | 68.4 | 74.1 | 63.8 |
| CoMB $_{Mix}$ | Book Corpus | 70.1 | 56.1 | 59.7 | 68.8 | 73.7 | 64.6 |
| CoMB $_{Mix + PC\ removed}$ | | 70.6 | 57.9 | **64.2** | **70.3** | 73.1 | **66.4** |
| *(c) Other unsupervised methods* | | | | | | | |
| Skip-thought[†] | Book Corpus | NA | 30.8 | 24.8 | 31.4 | 31.0 | 29.5 |
| WME | Google News | NA | **60.6** | 54.5 | 65.5 | 61.8 | 60.6 |
| SIF $_{PC\ removed}$ | Common Crawl | NA | 56.2 | 56.6 | 68.5 | 71.7 | 63.3 |
| *(d) Supervised methods* | | | | | | | |
| GenSen[†] | Multiple* | 66.4 | **60.6** | 54.7 | 65.8 | **74.2** | 63.8 |
| InferSent | AllNLI | **71.5** | 59.2 | 58.9 | 69.6 | 71.3 | 64.8 |

Table 1: Performance of CoMB and other baselines on the STS tasks. The numbers are average *Pearson correlation x 100*. Results are grouped into unsupervised (a, b, c) and supervised (d) methods. A further distinction is made between unsupervised results to indicate the source of word embeddings. ([†]) Skip-thought and GenSen scores are taken from Arora et al. (2017) and Kiros & Chan (2018) respectively. (*) GenSen uses a combination of multiple datasets: AllNLI, Book Corpus, WMT, etc. *'Mix'* denotes the mixed distributional estimate. *'PC removed'* refers to removing contribution along the principal component of point estimates as done in SIF. Detailed results of individual tasks can be found in S3. The best results overall are in **bold** while best results in a group are underlined.

the vanilla CoMB is better then SIF $_{PC\ removed}$ on average across the test set. Next, using the mixed distributional estimate (CoMB $_{Mix}$) improves the average test performance by 10%, and interestingly this is for mixing weight 0.4 (towards the point estimate). Further, when the PC removal is carried out for point estimates during mixing (i.e., CoMB $_{Mix + PC\ removed}$), the average performance goes to 57.7 and results in 18% relative improvement over SIF $_{PC\ removed}$.

**Ground Metric: Sent2Vec.** In parts (b, c, d) of Table 1, we see the effect of using an improved ground metric, by employing the word vectors obtained from Sent2vec[7]. Here, we notice that CoMB$_{Mix + PC\ removed}$ results in a performance of 66.4 and thus a relative improvement of 4% over Sent2vec's score of 63.8. This is a decent gain considering that for unstructured text corpora, Sent2vec is a state-of-the-art unsupervised method. Next, WME performs much worse than CoMB, showing the benefit of defining transport over contexts than words. Further, CoMB also outperforms popular supervised sentence embedding methods[8] such as GenSen and InferSent which utilize labeled corpora.

**Ablation studies and Qualitative analysis.** We perform an extensive ablation and qualitative analysis for sentence similarity. Due to space constraints, we enumerate the main observations here and details can be found in the supplementary section as follows. (a) Section S4.3: CoMB and SIF appear complementary in the nature of errors they make. CoMB outperforms when the difference in sentences stems from predicate while SIF is better when the distinguishing factor is the subject of the sentences. This is likely the reason why mixed distributional estimate helps in practice. (b) Section S2.3: we observe that by around $K = 300$ to $500$, the performance gained by increasing the number

---

[7]Sent2vec learns word embeddings so that their average works well as a sentence representation. We use the pre-trained embeddings available at `https://github.com/epfml/sent2vec`.

[8]USE (Cer et al., 2018), which relies on a labeled corpus, doesn't report results on STS12-15 but going by `https://github.com/google-research/bert/issues/128#issuecomment-451896503`, its performance is 67.5 which is close to CoMB's unsupervised performance of 66.4.

of clusters starts to plateau, implying that it is sufficient to only consider the representative contexts. (c) Section S4.4: CoMB generally fares better than SIF on datasets with longer sentences.

**Summary and further prospects.** We observe that using CoMB along with either GloVe or Sent2vec leads to a substantial boost, even taking beyond the performance of popular supervised methods such as GenSen and InferSent. Starting from the raw co-occurrence information, it takes **less than 11 minutes** to get all the STS results and see S1.4 for details. A future avenue would be to utilize the important property of non-associativity for Wasserstein barycenters (i.e., $B_p(\mu, B_p(\nu, \xi)) \neq B_p(B_p(\mu, \nu), \xi)$). This implies that we can take into account the word order with various aggregation strategies, like parse trees, to build the sentence representation by recursively computing barycenters phrase by phrase. However, this remains beyond the scope of this paper. Overall, this highlights the advantage of having distributional estimates for words, that can be extended to give a meaningful representation of sentences via CoMB in a principled manner.

# 7 HYPERNYMY DETECTION

In linguistics, hypernymy is a relation between words (or sentences) where the semantics of one word (the *hyponym*) are contained within that of another word (the *hypernym*). A simple form of this relation is the *is-A* relation, e.g., *cat* is an *animal*. Hypernymy is a special case of the more general concept of lexical entailment, the detection of which is relevant for tasks such as Question Answering (QA).

The early unsupervised approaches for this task exploited different linguistic properties of hypernymy (Weeds & Weir, 2003; Kotlerman et al., 2010; Santus et al., 2014; Rimell, 2014). While most of these are count-based, point embedding based methods (Chang et al., 2017; Henderson & Popa, 2016) have become more popular in recent years. Other approaches represent words by Gaussian distributions with KL-divergence as a measure of entailment (Vilnis & McCallum, 2014; Athiwaratkun & Wilson, 2017). These methods have proven to be powerful, as they not only capture the semantics but also the uncertainty about the contexts in which the word appears.

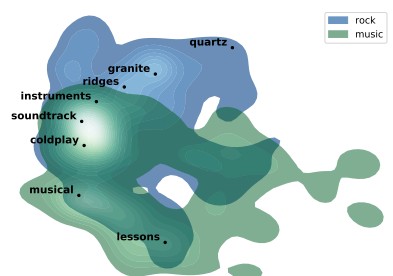

Figure 4: Distributional estimates of *rock* and *music*. The two words have an overlapping mode (for rock in the sense of rock music) and separate modes for other senses (such as rock in the sense of a stone).

Therefore, hypernymy detection is a great testbed to verify the effectiveness of our approach to represent each entity by the distribution of its contexts. The intuitive idea for the applicability of our method to this task originates from the *Distributional Inclusion Hypothesis* (Geffet & Dagan, 2005), which states that a word $v$ entails another word $w$ if "the most characteristic contexts of $v$ are expected to be included in all $w$'s contexts (but not necessarily amongst the most characteristic ones for $w$)". The inclusion of the contexts for the words *rock* and *music* is illustrated in Figure 4. We see our method as a relaxation of this strict inclusion condition, as a suitable entailment based ground metric in combination with CMD can more flexibly model this condition. Hence, it is natural to make use of the Context Mover's Distance (CMD), Eq. (5), but with an appropriate ground cost that measures entailment relations well.

For this purpose, we utilize a recently proposed method by (Henderson & Popa, 2016; Henderson, 2017) which explicitly models what information is known about a word, by interpreting each entry of the embedding as the degree to which a certain feature is present. Based on the logical definition of entailment they derive an operator measuring the entailment similarity between two so-called entailment vectors defined as follows: $\boldsymbol{v}_i \oslash \boldsymbol{v}_j = \sigma(-\boldsymbol{v}_i) \cdot \log \sigma(-\boldsymbol{v}_j)$, where the sigmoid $\sigma$ and $\log$ are applied component-wise on the embeddings $\boldsymbol{v}_i, \boldsymbol{v}_j$. Thus, we use as ground cost $D_{ij}^{\text{Hend.}} := -\boldsymbol{v}_i \oslash \boldsymbol{v}_j$. This asymmetric ground cost also shows that our framework can be flexibly used with an arbitrary cost function defined on the ground space.

**Evaluation.** In total, we evaluate our method on 10 standard datasets: BLESS (Baroni & Lenci, 2011), EVALution (Santus et al., 2015), (Benotto, 2015), (Weeds et al., 2014), BIBLESS (Kiela et al.,

| Method | Validation Set | Test Set | | | | | | |
| --- | --- | --- | --- | --- | --- | --- | --- | --- |
| | HypeNet-Train | HypeNet-Test | EVALution | LenciBenotto | Weeds | Turney | Baroni | BIBLESS |
| $D^{\text{Hend.}}$ | 29.0 | 28.8 | 31.6 | 44.8 | 60.8 | 56.6 | 78.3 | 70.5 |
| $\text{CMD}_{K=200}+D^{\text{Hend.}}$ | 53.4 | 53.4 | **38.1** | 50.1 | 63.9 | 56.0 | 67.5 | 74.0 |
| $\text{CMD}_{K=250}+D^{\text{Hend.}}$ | **53.6** | **53.7** | 37.1 | 49.9 | 63.8 | 56.3 | 67.3 | **74.9** |
| GE + C | NA | 21.6 | 26.7 | 43.3 | 52.0 | 53.9 | 69.7 | NA |
| GE + KL | NA | 23.7 | 29.6 | 45.1 | 51.3 | 52.0 | 64.6 | NA |
| DIVE + C·$\Delta$S | NA | 32.0 | 33.0 | **50.4** | **65.5** | **57.2** | **83.5** | NA |

Table 2: Comparison of the entailment vectors from (Henderson, 2017) used alone ($D^{\text{Hend.}}$), and when used together with our Context Mover's Distance ($\text{CMD}_K$) and $D^{\text{Hend.}}_{ij}$ as the underlying ground metric. The two listed CMD variants are the ones with best validation performance for $K = 200$ and $K = 250$ clusters. For reference, this table also includes state-of-the-art methods, like Gaussian embeddings with cosine similarity (GE+C) or KL-divergence (GE+KL), and DIVE.[10] The scores are **AP@all (%)**. More details about the training setup and results on other datasets can be found in Section S1.1, and Table S11 in Section S6.1. Best results are in **bold** and 2nd best are underlined.

2015), (Baroni et al., 2012), (Kotlerman et al., 2010), (Levy et al., 2014), HypeNet-Test (Shwartz et al., 2016), and (Turney & Mohammad, 2015). As an evaluation metric, we use average precision AP@all (Zhu, 2004). Following (Chang et al., 2017) we pushed any OOV (out-of-vocabulary) words in the test data to the bottom of the list, effectively assuming that the word pairs do not have a hypernym relation.

The foremost thing that we would like to check is the benefit of having a distributional estimate in comparison to just the point embeddings. Here, we observe that employing CMD along with the entailment embeddings, leads to a significant boost on most of the datasets, except on Baroni and Turney, where the performance is still competitive with the other state of the art methods like Gaussian embeddings. The more interesting observation is that on some datasets (EVALution, HypeNet, LenciBenotto) we even outperform or match state-of-the-art performance (cf. Table 2), by simply using CMD together with this ground cost $D^{\text{Hend.}}_{ij}$ based on the entailment embeddings. Notably, this approach is not specific to the entailment vectors from (Henderson, 2017) and more accurate set of vectors might help additionally. Alternatively, this also suggests that using CMD along with a method that produces embedding vectors (specialized for measuring the degree of entailment) can be a potential way to further improve the performance of that method. Some qualitative results taken from the BIBLESS dataset are listed in Table 3.

| Hyponym | Hypernym | Ground Truth | CMD | Henderson |
| --- | --- | --- | --- | --- |
| saw | tool | True | ✓ | ✗ |
| guitar | trumpet | False | ✓ | ✗ |
| battleship | vehicle | True | ✓ | ✗ |
| box | mortality | False | ✗ | ✓ |

Table 3: Excerpts from hypernymy detection on BIBLESS. For more examples and a detailed description, please refer to Tables S14/S15.

## 8 CONCLUSION

We advocate for representing entities by a distributional estimate on top of any given co-occurrence structure. For each entity, we jointly consider the histogram information (with its contexts) as well as the point embeddings of the contexts. We show how this enables the use of optimal transport over distributions of contexts. Our framework results in an efficient, interpretable and compositional metric to represent and compare entities (e.g. words) and groups thereof (e.g. sentences), while leveraging existing point embeddings. We demonstrate its performance on several NLP tasks such as sentence similarity and word entailment detection. Thus, a practical take-home message is: *do not throw away the co-occurrence information* (e.g. when using GloVe), but *instead pass it on to our method.* Motivated by the promising empirical results, applying the proposed framework on co-occurrence structures beyond NLP is an exciting direction.

---

[10]Scores for GE+C, GE+KL, and DIVE + C·$\Delta S$ are taken from (Chang et al., 2017) as we use the same evaluation setup. All the methods use a Wikipedia dump as a training corpus. In particular GE and DIVE employ WaCkypedia (a 2009 Wikipedia dump) from Baroni et al. (2009), and $D^{\text{Hend.}}$ and CMD are based on a 2015 Wikipedia dump.

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

# Appendices

In these appendices, we provide supplementary details on the experiments, mathematical framework, and detailed results in Section S1. In Section S2 we discuss computational aspects and the importance of clustering the contexts. Detailed results of the sentence representation and hypernymy detection experiments are listed on the following pages in Section S3 and S6 respectively. Then we describe a qualitative analysis of sentence similarity in Section S4, and finally discuss a qualitative analysis of hypernymy detection in Section S7.

## S1 TECHNICAL SPECIFICATIONS

In this Section, we give further details on the experimental framework in Section S1.1, on the PPMI formulation (Section S1.2), and on Optimal Transport (Section S1.3). In Section S1.5, we provide references for software release.

### S1.1 EXPERIMENTAL DETAILS

**Sentence Representations.**   While using the Toronto Book Corpus, we remove the errors caused by crawling and pre-process the corpus by filtering out sentences longer than 300 words, thereby removing a very small portion (500 sentences out of the 70 million sentences). We utilize the code[S1] from GloVe for building the vocabulary of size 205513 (obtained by setting min_count=10) and the co-occurrence matrix (considering a symmetric window of size 10). Note that as in GloVe, the contribution from a context word is inversely weighted by the distance to the target word, while computing the co-occurrence. The vectors obtained via GloVe have 300 dimensions and were trained for 75 iterations at a learning rate of 0.005, other parameters being the default ones. The performance of these vectors from GloVe was verified on standard word similarity tasks.

**Hypernymy Detection.**   The training of the entailment vector is performed on a Wikipedia dump from 2015 with 1.7B tokens that have been tokenized using the Stanford NLP library (Manning et al., 2014). In our experiments, we use a vocabulary with a size of 80'000 and word embeddings with 200 dimensions. We followed the same training procedure as described in Henderson (2017) and were able to reproduce their scores on the hypernymy detection task. For tuning the hyperparameters, we utilize the HypeNet training set of Shwartz et al. (2016) (from the random split), following the procedure indicated in Chang et al. (2017) for tuning DIVE and Gaussian embeddings.

### S1.2 PPMI DETAILS

**Formulation and Variants.**   Typically, the probabilities used in PMIare estimated from the co-occurrence counts $\#(w, c)$ in the corpus and lead to

$$\text{PPMI}(w, c) = \max\left(\log\left(\frac{\#(w, c) \times |Z|}{\#(w) \times \#(c)}\right), 0\right), \tag{8}$$

where, $\#(w) = \sum_c \#(w, c)$, $\#(c) = \sum_w \#(w, c)$ and $|Z| = \sum_w \sum_c \#(w, c)$. Also, it is known that PPMI is biased towards infrequent words and assigns them a higher value. A common solution is to smoothen[S2] the context probabilities by raising them to an exponent of $\alpha$ lying between $0$ and $1$. Levy & Goldberg (2014b) have also suggested the use of the shifted PPMI (SPPMI) matrix where the shift[S3] by $\log(s)$ acts like a prior on the probability of co-occurrence of target and context pairs.

---

[S1]https://github.com/stanfordnlp/GloVe

[S2]$p_\alpha(c) := \frac{\#(c)^\alpha}{\sum_{c'} \#(c')^\alpha}$.

[S3]Here, we denote the shift parameter by $s$ instead of the $k$ defined in (Levy et al., 2015) to avoid confusion with the other usage of $k$.

These variants of PPMI enable us to extract better semantic associations from the co-occurrence matrix. Finally, we have

$$\mathrm{SPPMI}_{\alpha,s}(w,c) := \max\left(\log\left(\frac{\#(w,c) \times \sum_{c'} \#(c')^\alpha}{\#(w) \times \#(c)^\alpha}\right) - \log(s), 0\right).$$

**Computational aspect.** We utilize the sparse matrix support of Scipy[S4] for efficiently carrying out all the PPMI computations.

**PPMI Column Normalizations.** In certain cases, when the PPMI contributions towards the partitions (or clusters) have a large variance, it can be helpful to consider the fraction of $\mathcal{C}_k$'s SPPMI (Eq. (9), (10)) that has been used towards a word $w$, instead of aggregate values used in (13). Otherwise the process of making the histogram unit sum might misrepresent the actual underlying contribution. We call this PPMI column normalization ($\beta$). In other words, the intuition is that the normalization will balance the effect of a possible non-uniform spread in total PPMI across the clusters. We observe that setting $\beta$ to 0.5 or 1 help in boosting performance on the STS tasks. The basic form of column normalization is shown in (10).

$$(\tilde{\mathbf{H}}^w)_k := \frac{(\bar{\mathbf{H}}^w)_k}{\sum_{k=1}^{K} (\bar{\mathbf{H}}^w)_k} \quad \text{with} \tag{9}$$

$$(\bar{\mathbf{H}}^w)_k := \frac{\mathrm{SPPMI}_{\alpha,s}(w,\mathcal{C}_k)}{\sum_w \mathrm{SPPMI}_{\alpha,s}(w,\mathcal{C}_k)}. \tag{10}$$

Another possibility while considering the normalization to have an associated parameter $\beta$ that can interpolate between the above normalization and normalization with respect to cluster size.

$$(\tilde{\mathbf{H}}_\beta^w)_k := \frac{(\bar{\mathbf{H}}_\beta^w)_k}{\sum_{k=1}^{K} (\bar{\mathbf{H}}_\beta^w)_k}, \quad \text{where}$$

$$(\bar{\mathbf{H}}_\beta^w)_k := \frac{\mathrm{SPPMI}_{\alpha,s}(w,\mathcal{C}_k)}{\sum_w \mathrm{SPPMI}_{\alpha,s}(w,\mathcal{C}_k)^\beta} \tag{11}$$

In particular, when $\beta = 1$, we recover the equation for histograms as in (10), and $\beta = 0$ would imply normalization with respect to cluster sizes.

## S1.3 OPTIMAL TRANSPORT

**Implementation aspects.** We make use of the Python Optimal Transport (POT)[S5] for performing the computation of Wasserstein distances and barycenters on CPU. For more efficient GPU implementation, we built custom implementation using PyTorch. We also implement a batched version for barycenter computation, which to the best of our knowledge has not been done in the past. The batched barycenter computation relies on a viewing computations in the form of block-diagonal matrices. As an example, this batched mode can compute around 200 barycenters in 0.09 seconds, where each barycenter is of 50 histograms (of size 100) and usually gives a speedup of about 10x.

**Scalability.** For further scalability, an alternative is to consider *stochastic optimal transport* techniques (Genevay et al., 2016). Here, the idea would be to randomly sample a subset of contexts from the distributional estimate while considering this transport.

**Stability of Sinkhorn Iterations.** For all our computations involving optimal transport, we typically use $\lambda$ around 0.1 and make use of log or median normalization as common in POT to stabilize the Sinkhorn iterations. Also, we observe that clipping the ground metric matrix (if it exceeds a particular large threshold) also sometimes results in performance gains.

---

[S4]https://docs.scipy.org/doc/scipy/reference/sparse.html
[S5]http://pot.readthedocs.io/en/stable/

**Value of $p$.** It has been shown in Agueh & Carlier (2011) that when the underlying space is Euclidean and $p = 2$, there exists a unique minimizer to the Wasserstein barycenter problem. But, since we are anyways solving the regularized Wasserstein barycenter (Cuturi & Doucet, 2014) problem over here instead of the exact one, the particular value of $p$ seems less of an issue. Empirically in the sentence similarity experiments, we have observed $p = 1$ to perform better than $p = 2$ (by about 2-3 points).

## S1.4 EMPIRICAL RUNTIME

Starting from scratch, it takes less than 11 minutes to get the results on all STS tasks which contains 25,000 sentences. This includes about 3 minutes to cluster 200,000 words (1 GPU), 5 minutes to convert raw co-occurrences into histograms of size 300 (1 CPU core) and 3 minutes for STS (1 GPU).

## S1.5 SOFTWARE RELEASE

**Core code and histograms.** Our code to build the ppmi-matrix, clusters, histograms as well computing Wasserstein distances and barycenters is publicly available on Github under `https://github.com/context-mover`. Precomputed histograms, clusters and point embeddings used in our experiments can also be downloaded from `https://drive.google.com/open?id=13stRuUd--71hcOq92yWUF-0iY15DYKNf`.

**Standard evaluation suite for Hypernymy.** To ease the evaluation pipeline, we have collected the most common benchmark datasets and compiled the code for assessing a model's performance on hypernymy detection or directionality into a Python package, called **HypEval**, which is publicly available at `https://github.com/context-mover/HypEval`. This also handles OOV (out-of-vocabulary) pairs in a standardized manner and allows for efficient, batched evaluation on GPU.

## S2 CLUSTERING THE CONTEXTS

In this Section, we discuss computational aspects and how using clustering makes the problem scalable. We give precise definition of the distributional estimate in Section S2.1, and show how the number of clusters affects the performance in Section S2.3.

## S2.1 COMPUTATIONAL CONSIDERATIONS.

The view of optimal transport between histograms of contexts introduced in Eq. (5) offers a pleasing interpretation (see Figure 2). However, it might be computationally intractable in its current formulation, since the number of possible contexts can be as large as the size of vocabulary (if the contexts are just single words) or even exponential (if contexts are considered to be phrases, sentences and otherwise). For instance, even with the use of SPPMI matrix, which also helps to sparsify the co-occurrences, the cardinality of the support of histograms still varies from $10^3$ to $5 \times 10^4$ context words, when considering a vocabulary of size around $2 \times 10^5$.

This is problematic because the Sinkhorn algorithm for regularized optimal transport (Cuturi, 2013, see Section 3) scales roughly quadratically in the histogram size, and the ground cost matrix can also become prohibitive to store in memory. One possible fix is to instead consider a set of representative contexts in this ground space, for example via clustering. We believe that with dense low-dimensional embeddings and a meaningful metric between them, we may not require as many contexts as needed before. For instance, this can be achieved by clustering the contexts with respect to metric $D_{\mathcal{G}}$. Apart from the computational gain, the clustering will lead to transport between more abstract contexts. This will although come at the loss of some interpretability.

Now, consider that we have obtained $K$ representative contexts, each covering some part $\mathcal{C}_k$ of the set of contexts $\mathcal{C}$. The histogram for word $w$ with respect to these contexts can then be written as:

$$\tilde{\mathbb{P}}_{\tilde{V}}^w = \sum_{k=1}^{K} (\tilde{\mathbf{H}}^w)_k \, \delta(\tilde{\mathbf{v}}_k). \tag{12}$$

Here $\tilde{\mathbf{v}}_k \in \tilde{V}$ is the point estimate of the $k^{th}$ representative context, and $(\tilde{\mathbf{H}}^w)_k$ denote the new histogram bin values with respect to the part $\mathcal{C}_k$,

$$(\tilde{\mathbf{H}}^w)_k := \frac{\text{SPPMI}_{\alpha,s}(w, \mathcal{C}_k)}{\sum_{k=1}^{K} \text{SPPMI}_{\alpha,s}(w, \mathcal{C}_k)}, \text{with} \tag{13}$$

$$\text{SPPMI}_{\alpha,s}(w, \mathcal{C}_k) := \sum_{c \in \mathcal{C}_k} \text{SPPMI}_{\alpha,s}(w, c). \tag{14}$$

In the following subsection, we show the effect of the number of clusters on the performance.

## S2.2 IMPLEMENTATION.

For clustering, we make use of kmcuda's[S6] efficient implementation of K-Means algorithm on GPUs.

## S2.3 EFFECT OF NUMBER OF CLUSTERS

Here, we analyze the impact of number of clusters on the performance of Context Mover's Barycenters (CoMB) for the sentence similarity experiments (cf. Section 6). In particular, we look at the three best performing variants (A, B, C) on the validation set (STS 16) as well as averaged across them.

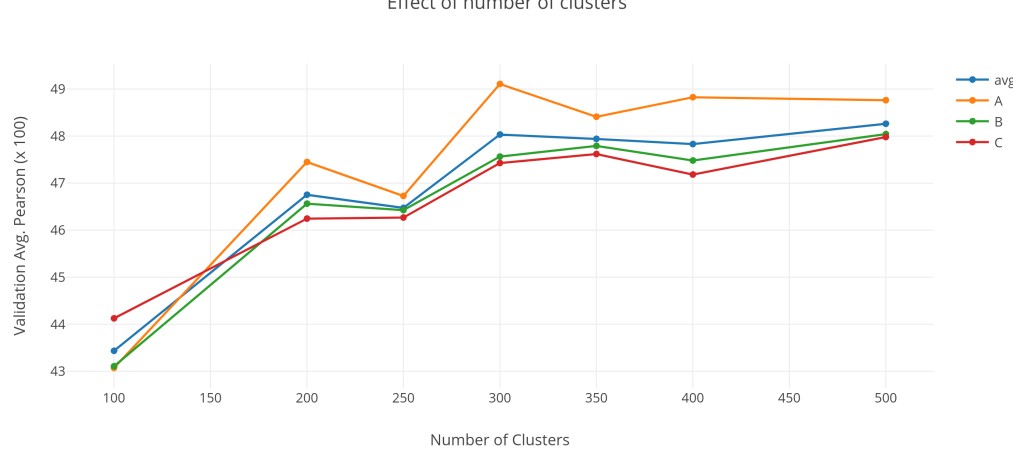

Figure S1: Effect of the number of clusters ($K$) on validation performance. A, B, C correspond to the three best performing variants of CoMB obtained as per validation on STS16 and as presented in the Table 1. In particular, A denotes the hyperparameter setting of [($\alpha$=0.55, $\beta$=1, $s$=5], B refers to [$\alpha$=0.55, $\beta$=0.5, $s$=5] and C denotes [$\alpha$=0.15, $\beta$=0.5, $s$=1]. The *'avg'* plot shows the average trend across these three configurations.

We observe in Figure S1 that on average the performance significantly improves when the number of clusters are increased until around $K = 300$, and beyond that mostly plateaus ($\pm$ 0.5). But, as can be seen for variants B and C the performance typically continues to rise until $K = 500$. It seems that the amount of PPMI column normalization ($\beta = 0.5$ vs $\beta = 1$) might be at play here.

As going from $K = 300$ to $K = 500$ comes at the cost of increased computation time, and doesn't lead to a substantial gain in performance. We use either $K = 300$ or $500$ clusters, depending on validation results, for our results on sentence similarity tasks.

Such a trend seems to be in line with the ideal case where we wouldn't need to do any clustering and just take all possible contexts into account. Thus, it suggests that better ways (other than clustering) to deal with this problem might further boost the performance.

---

[S6]https://github.com/src-d/kmcuda

## S3    Sentence Representation

We provide detailed results of the **test set performance** of *Context Mover's Barycenters* (CoMB) and related baselines on the STS-12, 13, 14 and STS-15 tasks in Tables S1 and S2 and **validation set performance** in Table S3.

| Model | STS12 | | | | | |
|---|---|---|---|---|---|---|
| | MSRpar | MSRvid | SMTeuroparl | WordNet | SMTnews | Average |
| NBoW | 17.5 | -6.4 | 25.4 | 37.2 | 31.9 | 21.1 |
| SIF | 12.1 | 51.6 | 23.5 | 55.1 | 19.9 | 32.4 |
| SIF + PC removed | 21.9 | 58.9 | 30.9 | 55.9 | 37.2 | 41.0 |
| CoMB (GloVe) | 31.3 | 61.5 | **47.5** | 54.5 | **46.0** | 48.2 |
| CoMB (GloVe) + Mix | **35.8** | 75.0 | 44.2 | 59.2 | 38.5 | **50.5** |
| CoMB (GloVe) + Mix + PC removed | 35.5 | **78.2** | 35.5 | **60.9** | 36.5 | 49.3 |
| Sent2vec | 37.7 | 78.7 | 49.3 | 70.2 | 42.3 | 55.6 |
| CoMB (sent2vec) + Mix | 40.7 | 78.9 | **49.9** | 68.0 | 43.0 | 56.1 |
| CoMB (sent2vec) + Mix + PC removed | **44.3** | **82.3** | 47.1 | **68.8** | **47.0** | **57.9** |

| Model | STS13 | | | |
|---|---|---|---|---|
| | FNWN | Headlines | WordNet | Average |
| NBoW | 14.2 | 27.1 | -0.8 | 13.5 |
| SIF | 8.5 | 54.1 | 6.3 | 23.0 |
| SIF + PC removed | 13.7 | 61.0 | 75.5 | 50.1 |
| CoMB (GloVe) | 11.8 | 54.6 | 60.1 | 42.2 |
| CoMB (GloVe) + Mix | 22.3 | 58.5 | 72.3 | 51.0 |
| CoMB (GloVe) + Mix + PC removed | **28.9** | **62.8** | **77.7** | **56.5** |
| Sent2vec | 42.4 | 66.2 | 62.7 | 57.1 |
| CoMB (sent2vec) + Mix | 42.5 | 67.6 | 69.1 | 59.7 |
| CoMB (sent2vec) + Mix + PC removed | **43.3** | **69.4** | **80.0** | **64.2** |

| Model | STS14 | | | | | | |
|---|---|---|---|---|---|---|---|
| | Forum | News | Headlines | Images | WordNet | Twitter | Average |
| NBoW | 18.2 | 37.6 | 24.0 | 14.9 | 17.1 | 38.0 | 25.0 |
| SIF | 21.1 | 29.4 | 50.7 | 34.3 | 22.4 | 46.5 | 34.1 |
| SIF + PC removed | 27.9 | 43.1 | 57.0 | 52.9 | 76.8 | 53.5 | 51.9 |
| CoMB (GloVe) | 40.4 | **64.9** | 50.5 | 51.5 | 64.4 | 57.8 | 54.9 |
| CoMB (GloVe) + Mix | **40.9** | 62.7 | 53.9 | 59.7 | 73.7 | 58.8 | 58.3 |
| CoMB (GloVe) + Mix + PC removed | 40.0 | 60.8 | **58.6** | **66.6** | **77.9** | **60.8** | **60.8** |
| Sent2vec | 49.1 | 67.2 | 63.9 | **82.5** | 72.4 | **75.5** | 68.4 |
| CoMB (sent2vec) + Mix | 52.1 | **69.5** | 63.2 | 78.3 | 75.1 | 74.5 | 68.8 |
| CoMB (sent2vec) + Mix + PC removed | **52.5** | 69.5 | **64.4** | 78.4 | **81.6** | 75.3 | **70.3** |

Table S1: Detailed **test set performance** of *Context Mover's Barycenters* (CoMB) and related baselines on the STS12, STS13, and STS14 tasks using Toronto Book Corpus. The numbers are average Pearson correlation x100 (with respect to groundtruth scores). 'Mix' denotes the mixed distributional estimate. 'PC removed' refers to removing contribution along the principal component of point estimates as done in SIF. The part in brackets after CoMB refers to the underlying ground metric.

The first 3 baselines (NBoW, and SIF twice) as well as the first three CoMB (first part of the Tables) are using Glove embeddings, while the last three methods (sent2vec and CoMB twice) use Sent2vec

| Model | STS15 | | | | | |
| | Forum | Students | Belief | Headlines | Images | Average |
|---|---|---|---|---|---|---|
| NBoW | 18.6 | 43.7 | 28.5 | 37.1 | 25.8 | 30.7 |
| SIF | 23.9 | 33.8 | 30.2 | 57.6 | 31.1 | 35.3 |
| SIF + PC removed | 35.3 | 63.8 | 51.0 | 62.3 | 51.6 | 52.8 |
| CoMB (GloVe) | 36.2 | _64.5_ | 45.2 | 61.1 | 61.8 | 53.8 |
| CoMB (GloVe) + Mix | _51.0_ | **66.2** | _54.4_ | _62.5_ | _68.2_ | _60.5_ |
| CoMB (GloVe) + Mix + PC removed | **55.3** | 61.3 | **63.3** | **66.1** | **74.1** | **64.0** |
| Sent2vec | 67.5 | **73.9** | **77.1** | 69.4 | **82.6** | **74.1** |
| CoMB (sent2vec) + Mix | _67.9_ | _73.8_ | _75.6_ | _69.8_ | 81.5 | _73.7_ |
| CoMB (sent2vec) + Mix + PC removed | **68.4** | 69.6 | 74.6 | **71.3** | _81.9_ | 73.1 |

Table S2: (continued from Table S1) Detailed **test set performance** of *Context Mover's Barycenters* (CoMB) and related baselines on the STS15 using Toronto Book Corpus. The numbers are average Pearson correlation x 100 (with respect to groundtruth scores). 'Mix' denotes the mixed distributional estimate. 'PC removed' refers to removing contribution along the principal component of point estimates as done in SIF. The part in brackets after CoMB refers to the underlying ground metric.

| Model | STS16 | | | | | |
| | Answer | Headlines | Plagiarism | Postediting | Question | Average |
|---|---|---|---|---|---|---|
| NBoW | 19.9 | 32.6 | 16.5 | 35.7 | -8.9 | 19.2 |
| SIF | 35.1 | 55.1 | 14.6 | 31.7 | -3.5 | 26.6 |
| SIF + PC removed | 42.4 | _60.0_ | 58.5 | _71.7_ | 55.4 | 57.6 |
| CoMB (GloVe) | 38.7 | 55.4 | 50.2 | 67.6 | 50.1 | 52.4 |
| CoMB (GloVe) + Mix | **50.5** | 57.1 | _64.2_ | 69.6 | _59.7_ | _60.2_ |
| CoMB (GloVe) + Mix + PC removed | _47.9_ | **60.6** | **70.0** | **76.6** | **59.9** | **63.0** |
| Sent2vec | 62.5 | 68.3 | **78.6** | 82.5 | 53.5 | 69.1 |
| CoMB (sent2vec) + Mix | _62.6_ | _69.0_ | _76.4_ | _83.0_ | _59.5_ | _70.1_ |
| CoMB (sent2vec) + Mix + PC removed | **63.3** | **69.7** | 74.8 | **83.9** | **61.1** | **70.6** |

Table S3: Detailed **validation set performance** of *Context Mover's Barycenters* (CoMB) and related baselines on the STS16 using Toronto Book Corpus. The numbers are average Pearson correlation x100 (with respect to groundtruth scores). 'Mix' denotes the mixed distributional estimate. 'PC removed' refers to removing contribution along the principal component of point estimates as done in SIF. The part in brackets after CoMB refers to the underlying ground metric. Note that, STS16 was used as the validation set to obtain the best hyperparameters for all the methods in these experiments. As a result, high performance on STS16 may not be indicative of the overall performance.

embeddings (second part of the Tables). The numbers are average Pearson correlation (with respect to ground-truth scores).

We observe empirically that the PPMI smoothing parameter $\alpha$, which balances the bias of PPMI towards rare words, plays an important role. While its ideal value would vary on each task, we found the settings mentioned in the Table S4 to work well uniformly across the above spectrum of tasks. We also provide in Table S4 a comparison of the hyper-parameters used in each of the methods in Tables S1, S2 and S3

| | $a$ | | | Clusters | PC removed | Mixing |
|---|---|---|---|---|---|---|
| SIF | $a = 10^{-4}$ | | | | | |
| SIF + PC removed | $a = 10^{-4}$ | | | | ✓ | 0.4 |
| | $\alpha$ | $\beta$ | $s$ | | | |
| CoMB (GloVe) | 0.55 | 1 | 5 | 300 | | |
| CoMB (GloVe) + Mix | 0.95 | 1 | 1 | 500 | | 0.4 |
| CoMB (GloVe) + Mix + PC removed | 0.95 | 1 | 1 | 500 | ✓ | 0.4 |
| CoMB (sent2vec) + Mix | 0.15 | 1 | 1 | 300 | | 0.4 |
| CoMB (sent2vec) + Mix + PC removed | 0.15 | 1 | 1 | 300 | ✓ | 0.4 |

Table S4: Detailed parameters for the methods presented in Tables S1, S1 and S3. The parameters for CoMB $\alpha, \beta, s$ denote the PPMI smoothing, column normalization exponent (Eq. (11)), and k-shift.

## S4 QUALITATIVE ANALYSIS OF SENTENCE SIMILARITY

In this section, we aim to qualitatively analyse the particular examples where our method, Context Mover's Barycenters (CoMB), performs better or worse than the Smooth Inverse Frequency (SIF) approach from Arora et al. (2017).

### S4.1 EVALUATION PROCEDURE

**Comparing by rank.** It doesn't make much sense to compare the raw distance values between two sentences as given by Context Mover's Distance (CMD) for CoMB and cosine distance for SIF. This is because the spread of distance values across sentence pairs can be quite different. Note that the quantitative evaluation of these tasks is also carried out by Pearson/Spearman rank correlation of the predicted distances/similarities with the ground-truth scores.

Thus, in accordance with this reasoning, we compare the similarity score of a sentence pair relative to its rank based on ground-truth score (amongst the sentence pairs for that dataset). So, the better method should rank sentence pairs closer to the ranking obtained via ground-truth scores.

| *Ground-Truth Score* | *Implied meaning* |
|---|---|
| 5 | The two sentences are completely equivalent, as they mean the same thing. |
| 4 | The two sentences are mostly equivalent, but some unimportant details differ. |
| 3 | The two sentences are roughly equivalent, but some important information differs/missing. |
| 2 | The two sentences are not equivalent, but share some details. |
| 1 | The two sentences are not equivalent, but are on the same topic. |
| 0 | The two sentences are completely dissimilar. |

Table S5: STS ground scores and their implied meanings, as taken from Agirre et al. (2015)

**Ground-truth details.** The ground-truth scores (can be fractional) and range from 0 to 5, and the meaning implied by the integral score values can be seen in the Table S5. In the case where different examples have the same ground-truth score, the ground-truth rank is then based on lexicographical ordering of sentences for our qualitative evaluation procedure. (This for instance means that sentence pairs ranging from 62 to 74 would correspond to the same ground-truth score of 4.6). The ranking is done in the descending order of sentence similarity, i.e., most similar to least similar.

**Example selection criteria.** For all the examples, we compare the best variants of CoMB and SIF on those datasets. We particularly choose those examples where there is the maximum difference in ranks according to CoMB and SIF, as they would be more indicative of where a method succeeds or fails. Nevertheless, such a qualitative evaluation is subjective and is meant to give a better understanding of things happening under the hood.

## S4.2 Experiments and Observations

We look at examples from three datasets, namely: Images from STS15, News from STS14 and WordNet from STS14 to get a better idea of an overall behavior. In terms of aggregate quantitative performance, on Images and News datasets, CoMB is better than SIF, while the opposite is true for WordNet. These examples across the three datasets may not probably be exhaustive and are up to subjective interpretation, but hopefully will lend some indication as to where and why each method works.

### S4.2.1 Task: STS14, Dataset: News

We look in detail at the examples in News dataset from STS 2014 (Agirre et al., 2014). The results of qualitative analysis on Images and WordNet datasets can be found in Section S4.5. For reference, CoMB results in a better performance overall with a Pearson correlation (x100) of 64.9 versus 43.0 for SIF, as presented in Table S1. The main observations are:

**Observation 1.** Examples 1, 2, 4, 5 are sentence pairs which are equivalent in meaning (cf. Table S5), but typically have additional details in the predicates of the sentences. Here, CoMB is better than SIF at ranking the pairs closer to the ground-truth ranking. This probably suggests the averaging of word embeddings, which is the $1^{st}$ step in SIF, is not as resilient to the presence of such details than the Wasserstein barycenter of distributional estimates in CoMB. We speculate that when having distributional estimates (where multiple senses or contexts are considered), adding details can help towards refining the particular meaning implied.

**Observation 2.** Let's consider the examples 3 and 6 where SIF is better than CoMB. These are sentence pairs which are equivalent or roughly equivalent in meanings, but with a few words substituted (typically subjects) like *"judicial order"* instead of *"court"* in example 3. Here it seems that the substitution is adverse for CoMB while considering varied senses through the distributional estimate, in comparison to looking at the "point" meaning given by SIF.

**Observation 3.** In 7, 8, and 10, each sentence pair is about a common topic, but the meaning of individual sentences is quite different. For instance, example 8: *"south korea launches new bullet train reaching 300 kph"* & *"south korea has had a bullet train system since the 1980s"*. Or like in example 10: *"china is north korea ' s closest ally"* & *"north korea is a reclusive state"*. Note that typically in these examples, the subject is same in a sentence pair, and the difference is mainly in the predicate. Here, CoMB identifies the difference and ranks them closer to the ground-truth. Whereas, SIF fails to understand this and ranks them as more similar (and far away) than the ground-truth.

**Observation 4.** The examples 9, 11, and 12 are related sentences and differ mainly in details such as the name of the country, person, department, i.e. proper nouns. In particular, consider example 9: *"south korea and israel oppose proliferation of weapons of mass destruction and an arms race"* & *"china will resolutely oppose the proliferation of mass destructive weapons"*. The main difference in these examples stems from differences in the subject rather than the predicate. CoMB considers these sentence pairs to be more similar than suggested by ground-truth. Hence, in such scenarios where the subject (like the particular proper nouns) makes the most difference, SIF seems to be better.

## S4.3 Conclusions from Qualitative Examples

Summarizing the observations from the above qualitative analysis on News dataset[S7], we conclude the following about the nature of success or failures of each method.

- When the subject of the sentence is similar and main difference stems from the predicate, CoMB is the winner. This can be seen for both the case when predicates are equivalent but described distinctly *(observation 1)* and when predicates are not equivalent *(observation 3)*.

- When the predicates are similar and the distinguishing factor is in the subject (or object), SIF takes the lead. This seems to be true for both scenarios when the subject used increases or decreases the similarity as measured by CoMB, *(observations 2 and 4)*.

---

[S7]Similar findings can also be seen for the two other datasets in Section S4.5.

| | Sentence 1 | Sentence 2 | Ground-Truth Score | Ground-Truth Ranking | CoMB Ranking | SIF Ranking |
|---|---|---|---|---|---|---|
| 1 | the united states government and other nato members have refused to ratify the amended treaty until officials in moscow withdraw troops from the former soviet republics of moldova and georgia . | the united states and other nato members have refused ratify the amended treaty until russia completely withdraws from moldova and georgia . | 4.6 | 30 | **67** | 152 |
| 2 | jewish-american group the anti-defamation league ( adl ) published full-page advertisements in swiss and international papers in april 2008 accusing switzerland of funding terrorism through the deal . | the anti-defamation league took out full-page advertisments in swiss and international newspapers earlier in april 2008 accusing switzerland of funding terrorism through the deal . | 4.4 | 36 | **35** | 128 |
| 3 | the judicial order accused raghad of funding terrorism . | the court accused raghad saddam hussein of funding terrorism . | 4.2 | 59 | 258 | **124** |
| 4 | estonian officials stated that some of the cyber attacks that caused estonian government websites to shut down temporarily came from computers in the administration of russia including in the office of president vladimir putin . | officials in estonia including prime minister andrus ansip have claimed that some of the cyber attacks came from russian government computers including computers in the office of russian president vladimir putin . | 3.8 | 86 | **84** | 206 |
| 5 | the african union has proposed a peace-keeping mission to help somalia ' s struggling transitional government stabilize somalia . | the african union has proposed a peace-keeping mission to aid the struggling transitional government in stabilizing somalia , particularly after the withdrawal of ethiopian forces | 3.6 | 119 | **104** | 262 |
| 6 | some asean officials stated such standardization would be difficult due to different countries ' political systems . | some officials stated the task would be difficult for asean members because of varied legal and political systems . | 3.6 | 117 | 244 | **108** |
| 7 | nicaragua commemorated the 25th anniversary of the sandinista revolution . | nicaragua has not reconciled how to approach the anniversary of the sandinista revolution . | 2.4 | 213 | **250** | 48 |
| 8 | south korea launches new bullet train reaching 300 kph . | south korea has had a bullet train system since the 1980s . | 2 | 232 | **267** | 130 |
| 9 | south korea and israel oppose proliferation of weapons of mass destruction and an arms race . | china will resolutely oppose the proliferation of mass destructive weapons . | 1.4 | 262 | 164 | **235** |
| 10 | china is north korea ' s closest ally . | north korea is a reclusive state . | 1.2 | 265 | **279** | 196 |
| 11 | the chinese government gave active co-operation and assistance to the organization for the prohibition of chemical weapons inspections . | the ecuadorian foreign ministry said in a statement that delegates from the organization for the prohibition of chemical weapons ( opaq ) will also take part in the meeting . | 1 | 277 | 158 | **231** |
| 12 | do quy doan is a spokesman for the vietnamese ministry of culture and information . | grenell is spokesman for the u.s. mission to the united nations . | 0.8 | 282 | 213 | **292** |

Table S6: Examples of some indicative sentence pairs, from *News* dataset in *STS14*, with ground-truth scores and ranking as obtained via (best variants of) CoMB and SIF. The total number of sentences is *300* and the ranking is done in descending order of similarity. The method which ranks an example closer to the ground-truth rank is better and is highlighted in **blue**. CoMB ranking is the one produced when representing sentences via CoMB and then using CMD to compare them. SIF ranking is when sentences are represented via SIF and then employing cosine similarity.

- The above two points in a way also signify where having distributional estimates can be better or worse than point estimates.
- CoMB and SIF appear to be complementary in the kind of errors they make. Hence, combining the two is an exciting future avenue.

Lastly, it also seems worthwhile to explore having different ground metrics for CoMB and CMD (which are currently shared). The ground metric plays a crucial role in performance and the nature of these observations. Employing a ground metric(s) that better handles the above subtleties would be a useful research direction.

### S4.4 EFFECT OF SENTENCE LENGTH

In this section, we look at the length of sentences across all the datasets in each of the STS tasks. Average sentence length is one measure of the complexity of a particular dataset. But looking at just sentence lengths may not give a complete picture, especially for the textual similarity tasks where there can be many words common between the sentence pairs. The Table S7 shows the various statistics of each dataset, with respect to the sentence lengths along with the better method on each of them (out of CoMB and SIF).

| Task-Dataset | # sentence pairs | Avg. sentence length | Avg. word overlap (per sentence pair) | Avg. effective sentence length (excluding common words) | Better method |
|---|---|---|---|---|---|
| STS12-MSRpar | 750 | 21.16 | 14.17 | 6.99 | CoMB |
| STS12-MSRvid | 750 | 7.65 | 4.70 | 2.95 | CoMB |
| STS12-SMTeuroparl | 459 | 12.33 | 8.11 | 4.22 | CoMB |
| STS12-WordNet | 750 | 8.82 | 5.03 | 3.79 | SIF |
| STS12-SMTnews | 399 | 13.62 | 8.66 | 4.96 | SIF |
| STS13-FNWN | 189 | 22.94 | 2.53 | 20.41 | CoMB |
| STS13-Headlines | 750 | 7.80 | 3.76 | 4.05 | SIF |
| STS13-WordNet | 561 | 8.17 | 4.64 | 3.53 | SIF |
| STS14-Forum | 450 | 10.48 | 7.03 | 3.45 | CoMB |
| STS14-News | 300 | 17.42 | 11.59 | 5.83 | CoMB |
| STS14-Headlines | 750 | 7.91 | 3.89 | 4.01 | SIF |
| STS14-Images | 750 | 10.18 | 6.20 | 3.98 | SIF |
| STS14-WordNet | 750 | 8.87 | 4.83 | 4.05 | SIF |
| STS14-Twitter | 750 | 12.25 | 4.85 | 7.40 | (equal) |
| STS15-Forum | 375 | 17.77 | 4.29 | 13.49 | CoMB |
| STS15-Students | 750 | 10.70 | 5.33 | 5.37 | CoMB |
| STS15-Belief | 375 | 16.53 | 6.27 | 10.26 | SIF |
| STS15-Headlines | 750 | 8.00 | 3.71 | 4.29 | SIF |
| STS15-Images | 750 | 10.66 | 6.07 | 4.59 | CoMB |

Table S7: Analysis of sentence lengths in each of the datasets from STS12, STS13, STS14, and STS15. Along with the average sentence lengths, we also measure average word overlap in the sentence pair and thus the average *effective sentence length (i.e., after excluding the overlapping/common words in the sentence pair)*. For reference, we also show which out of CoMB or SIF performs better. On STS14-Twitter, the difference in performance isn't significant and we thus write 'equal' in the corresponding cell.

**Observations.**

- We notice that on datasets with longer effective sentence lengths, CoMB performs better than SIF on average. There might be other factors at play here, but if one had to pick on the axis of effective sentence length, CoMB leads over SIF[S8].

---

[S8]Effective sentence length averaged across datasets where CoMB is better is **7.48**. Contrast this to an average effective sentence length of **5.03** across datasets where SIF is better.

- The above statement also aligns well with the *observation 1* from the qualitative analysis (cf. Section S4.2.1), that having more details can help in refining the particular meaning or sense implied by CoMB. (Effective sentence length can serve as a good proxy for indicating the amount of details.)

- It also seems to explain why both methods don't perform well (see Table S1) on STS13-FNWN, which has on average the maximum effective sentence length (of 20.4).

- To an extent, it also points towards the effect of corpora. For instance, in a corpus such as WordNet, which has a low average sentence length and with examples typically concerned about word definitions (see Table S9), SIF seems to be better of the methods. On the other hand, CoMB seems to be better for News (Table S6), Image captions (Table S8) or Forum.

## S4.5 Additional Qualitative Analysis

### S4.5.1 Task: STS15, Dataset: Images

We consider the sentence pairs from Images dataset in STS15 task (Agirre et al., 2015), as presented in Table S8. As a reminder, CoMB outperforms SIF on this dataset with a Pearson correlation (x100) of 61.8 versus 51.7, as mentioned in Table S2. The main observations are:

| | Sentence 1 | Sentence 2 | Ground-Truth Score | Ground-Truth Ranking | CoMB Ranking | SIF Ranking |
|---|---|---|---|---|---|---|
| 1 | the man and two young boys jump on a trampoline . | a man and two boys are bouncing on a trampoline . | 4.8 | 68 | **74** | 640 |
| 2 | a boy waves around a sparkler . | a young boy is twisting a sparkler around in the air . | 4.4 | 126 | **195** | 624 |
| 3 | a dog jumps in midair to catch a frisbee . | the brown dog jumps for a pink frisbee . | 4 | 184 | **161** | 481 |
| 4 | a child is walking from one picnic table to another . | the boy hops from one picnic table to the other in the park . | 3.2 | 287 | **401** | 737 |
| 5 | three boys are running on the beach playing a game . | two young boys and one young man run on a beach with water behind them . | 3.2 | 306 | **260** | 421 |
| 6 | a boy swinging on a swing . | the girl is on a swing . | 2.4 | 380 | **410** | 622 |
| 7 | a man is swinging on a rope above the water . | a man in warm clothes swinging on monkey bars at night . | 1.6 | 492 | 259 | **606** |
| 8 | a skier wearing blue snow pants is flying through the air near a jump . | a skier stands on his hands in the snow in front of a movie camera . | 1.4 | 514 | 264 | **605** |
| 9 | two black and white dogs are playing together outside . | two children and a black dog are playing out in the snow . | 1 | 570 | 185 | **372** |
| 10 | three dogs running in the dirt . | the yellow dog is running on the dirt road . | 1 | 524 | 303 | **531** |
| 11 | a little girl and a little boy hold hands on a shiny slide . | a little girl in a paisley dress runs across a sandy playground . | 0.4 | 629 | **683** | 354 |
| 12 | a little girl walks on a boardwalk with blue domes in the background . | a man going over a jump on his bike with a river in the background . | 0 | 696 | 310 | **591** |

Table S8: Examples of some indicative sentence pairs, from *Images* dataset in *STS15*, with ground-truth scores and ranking as obtained via (best variants of) CoMB and SIF. The total number of sentences is **750** and the ranking is done in descending order of similarity. The method which ranks an example closer to the ground-truth rank is better and is highlighted in **blue**. CoMB ranking is the one produced when representing sentences via CoMB and then using CMD to compare them. SIF ranking is when sentences are represented via SIF and then employing cosine similarity.

**Observation A.** Example 1 to 5 indicate pairs of sentences which are essentially equivalent in meaning, but with varying degrees of equivalence. Here, we can see that CoMB with CMD is able to rank the similarity between these pairs quite well in comparison to SIF, even when their way of describing is different. For instance, example 2 : *"a boy waves around a sparkler"* & *"a young boy is twisting a sparkler around in the air"*. This points towards the benefit of having multiple senses or contexts encoded through the distributional estimate in CoMB.

**Observation B.** Next, in the examples 7 to 10, which consist of sentence pairs that are not equivalent but have commonalities (about the topic). Here, SIF ranks the sentences closer to the ground-truth ranking while CoMB interprets these pairs as being more common in meaning than given by ground-truth. This could be the consequence of comparing the various senses or contexts implied by the sentence pairs via CMD. Take for instance, example 10, *"three dogs running in the dirt"* & *"the yellow dog is running on the dirt road"*. Since these sentences are about the similar topic (and the major difference is in their subject), this can result in CMD considering them more similar than cosine distance.

**Observation C.** For sentences which are completely dissimilar as per ground-truth, let's look at example 11 and 12. Consider 11, which is *"a little girl and a little boy hold hands on a shiny slide"* & *"a little girl in a paisley dress runs across a sandy playground"*, the sentences meaning totally different things and CoMB seems to be better at ranking than SIF. But, consider example 12: *"a little girl walks on a boardwalk with blue domes in the background"* & *"a man going over a jump on his bike with a river in the background"*. One common theme[S9] can be thought as *"a person moving with something blue in the background"*, which can result in CoMB ranking the sentence as more similar. SIF also ranks it higher (at 591) than ground-truth (696), but is more closer than CoMB which ranks it at 310.

### S4.5.2    TASK: STS14, DATASET: WORDNET

| | Sentence 1 | Sentence 2 | Ground-Truth Score | Ground-Truth Ranking | CoMB Ranking | SIF Ranking |
|---|---|---|---|---|---|---|
| 1 | combine so as to form a more complex product . | combine so as to form a whole ; mix . | 4.6 | 127 | **142** | 335 |
| 2 | ( cause to ) sully the good name and reputation of . | charge falsely or with malicious intent ; attack the good name and reputation of someone . | 4.4 | 176 | **235** | 534 |
| 3 | a person or thing in the role of being a replacement for something else | a person or thing that takes or can take the place of another . | 4.2 | 248 | **270** | 535 |
| 4 | create something in the mind . | form a mental image of something that is not present or that is not the case . | 3.6 | 340 | **443** | 683 |
| 5 | the act of surrendering an asset | the act of losing or surrendering something as a penalty for a mistake or fault or failure to perform etc . | 3 | 405 | **445** | 639 |
| 6 | ( attempt to ) convince to enroll , join or participate | register formally as a participant or member . | 2.8 | 406 | **423** | 507 |
| 7 | return to a prior state . | return to an original state . | 4.4 | 219 | 384 | **231** |
| 8 | give away something that is not needed . | give up what is not strictly needed . | 4.2 | 261 | 709 | **383** |
| 9 | a person who is a member of the senate . | a person who is a member of a partnership . | 0.4 | 553 | 260 | **429** |
| 10 | the context or setting in which something takes place . | the act of starting something . | 0 | 717 | 485 | **707** |
| 11 | a spatial terminus or farthest boundary of something . | a relation that provides the foundation for something . | 0 | 620 | 500 | **623** |
| 12 | the act of beginning something new . | the act of rejecting something . | 0 | 670 | **677** | 539 |

Table S9: Examples of some indicative sentence pairs, from *WordNet* dataset in *STS14*, with ground-truth scores and ranking as obtained via (best variants of) CoMB and SIF. The total number of sentences is **750** and the ranking is done in descending order of similarity. The method which ranks an example closer to the ground-truth rank is better and is highlighted in **blue**. CoMB ranking is the one produced when representing sentences via CoMB and then using CMD to compare them. SIF ranking is when sentences are represented via SIF and then employing cosine similarity.

Lastly, we discuss the examples and observations derived from the qualitative analysis on WordNet dataset from STS14 (Agirre et al., 2014). This dataset is comprised of sentences which are the definitions of words/phrases, and sentence length is typically smaller than the datasets discussed

---

[S9]Of course, this is upto subjective interpretation.

before. For reference, SIF (76.8) does better than CoMB (64.4) in terms of average Pearson correlation (x100), as mentioned in Table S1.

**Observation D.**  Consider examples 1 to 6 as shown in Table S9, which fall in the category of equivalent sentences but in varying degrees. The sentence pairs essentially indicate different ways of characterizing equivalent things. Here, CoMB is able to rank the similarity between sentences in a better manner than SIF. Specifically, see example 2: *"( cause to ) sully the good name and reputation of"* & *"charge falsely or with malicious intent ; attack the good name and reputation of someone"*. It seems that SIF is not able to properly handle the additional definition present in sentence 2 and ranks this pair much lower in similarity at 534 versus 235 for CoMB. This is also in line with observation 1 about added details in the Section S4.2.1.

**Observation E.**   In the examples 7 to 9, where CoMB doesn't do well in comparison to SIF, mainly have a slight difference in the object of the sentence. For instance, in example 9: *"a person who is a member of the senate"* & *"a person who is a member of a partnership"*. So based on the kind of substituted word, looking at its various contexts via the distributional estimate can make it more or less similar than desired. In such cases, using the "point" meanings of the objects seems to fare better. This also aligns with the observations 2 and 4 in the Section S4.2.1.

## S5  NEAREST NEIGHBOR ANALYSIS

Here, we would like to qualitatively probe the kind of results obtained when computing Wasserstein barycenter of the distributional estimates, in particular, when using CoMB to represent sentences. To this end, we consider a few simple sentences and find the closest word in the vocabulary for CoMB (with respect to CMD) and contrast it to SIF with cosine distance.

| Query | CoMB (with CMD) | SIF (with cosine, no PC removal) |
| --- | --- | --- |
| ['i', 'love', 'her'] | love, hope, *always*, *actually*, *because*, *doubt*, *imagine*, *but*, *never*, *simply* | love, loved, breep-breep, *want*, clash-clash-clang, thysel, *know*, think, nope, *life* |
| ['my', 'favorite', 'sport'] | sport, *costume*, circus, *costumes*, *outfits*, super, sports, *tennis*, *brand*, fabulous | favorite, favourite, sport, wiccan-type, *pastime*, pastimes, sports, best, *hangout*, spectator |
| ['best', 'day', 'of', 'my', 'life'] | best, *for*, *also*, only, or, *anymore*, *all*, *is*, *having*, *especially* | life, day, best, c.5, writer/mummy, days, margin-bottom, time, margin-left,night |
| ['he', 'lives, 'in', 'europe', 'for'] | america, europe, *decades*, asia, *millenium*, preserve, *masters*, *majority*, elsewhere, *commerce* | lives, europe, life, america, lived, world, england, france, people, c.5 |
| ['he', 'may', 'not', 'live'] | *unless*, *perhaps*, must, may, *anymore*, will, likely, youll, would, certainly | may, live, should, will, might, must, margin-left, henreeeee, 0618082132, think |
| ['can', 'you', 'help', me', 'shopping'] | *anytime*, *yesterday*, *skip*, *overnight*, *wed*, *afterward*, choosing, figuring, deciding, shopping | help, can, going, want, *go*, *do*, think, need, able, take |
| ['he', 'likes', 'to', 'sleep', 'a', 'lot'] | *whenever*, forgetting, *afterward*, *pretending*, rowan, eden, *casper*, nash, annabelle, savannah, | lot, sleep, much, *besides*, better, likes, *really*, think, *probably*, talk |

Table S10: Top 10 closest neighbors for CoMB and SIF (no PC removed) found across the vocabulary, and sorted in ascending order of distance from the query sentence. Words in *italics* are those which in our opinion would fit well when added to one of the places in the query sentence. Note that, both CoMB (under current formulation) and SIF don't take the word order into account.

**Observations.**   We find that closest neighbors (see Table S10) for CoMB consist of relatively more diverse set of words which fit well in the context of given sentence. For example, take the sentence "i love her", where CoMB captures a wide range of contexts, for example, "i *actually* love her", "i love her *because*", "i *doubt* her love" and more. Also for an ambiguous sentence "he lives in europe for", the obtained closest neighbors for CoMB include: 'decades', 'masters', 'majority', 'commerce' , etc., while with SIF the closest neighbors are mostly words similar to one of the query words. Further, if you look at the last three sentences in the Table S10, the first closest neighbor for CoMB even acts as a good next word for the given query. This suggests that CoMB might perform well on the task of sentence completion, but this additional evaluation is beyond the scope of this paper.

## S6 Hypernymy Detection

In this Section, we provide detailed results for the hypernymy detection in Section S6.1 and mention the corresponding hyperparamters in Section S6.2. We also mention the effect of PPMI parameters on Hypernymy results in Section S6.3.

### S6.1 Detailed Results

| | Dataset | | | | | |
|---|---|---|---|---|---|---|
| Method | BLESS | EVALution | LenciBenotto | Weeds | BIBLESS | Baroni |
| Henderson et al. ($D^{\text{Hend.}}$) | **6.4** | 31.6 | 44.8 | 60.8 | 70.5 | **78.3** |
| CMD ($K$=200) + $D^{\text{Hend.}}$ | 5.8 | **38.1** | **50.1** | **63.9** | 74.0 | **67.5** |
| CMD ($K$=250) + $D^{\text{Hend.}}$ | 5.8 | 37.1 | 49.9 | 63.8 | **74.9** | 67.3 |

| | Dataset | | | | |
|---|---|---|---|---|---|
| Method | Kotlerman | Levy | HypeNet-Test | Turney | **Avg.Gain** |
| Henderson et al. ($D^{\text{Hend.}}$) | 34.0 | 11.7 | 28.8 | **56.6** | - |
| CMD ($K$=200) + $D^{\text{Hend.}}$ | **34.7** | 12.2 | 53.4 | 56.0 | +3.2 |
| CMD ($K$=250) + $D^{\text{Hend.}}$ | 34.4 | **12.9** | **53.7** | 56.3 | **+ 3.3** |

Table S11: Comparison of the entailment vectors alone (Hend.) and when used together with our Context Mover's Distance, CMD($K$) (where $K$ is the number of clusters), in the form of ground cost $D^{\text{Hend.}}$. We also indicate the average gain in performance across these 10 datasets by using CMD along with the entailment vectors. All scores are AP at all (%).

### S6.2 Hyperparameters

The above listed variants of CMD are the ones with best validation performance on HypeNet-Train (Shwartz et al., 2016). The other hyperparameters (common) for both of them are as follows:

- PPMI smoothing, $\alpha = 0.5$.
- PPMI column normalization exponent, $\beta$=0.5.
- PPMI k-shift, $s$=1.
- Regularization constant for Wasserstein distance, $\lambda$=0.1
- Number of Sinkhorn iterations = 500.
- Log normalization of Ground Metric.

**Out of Vocabulary Details.** The following shows the out of vocabulary information for entailment experiments.

### S6.3 Effect of PPMI parameters for Hypernymy Detection

This table was generated during an earlier version of the paper, when we were not considering the validation on HypeNet-Train. Hence, the above table doesn't contain numbers on HypeNet-Test, but an indication of performance on it can be seen in Section S11. In any case, this table suggests that our method works well for several PPMI hyper-parameter configurations.

| Dataset | Number of pairs (N) | Out of vocabulary pairs (OOV) |
|---|---|---|
| BLESS | 26554 | 1504 |
| EVALution | 13675 | 92 |
| LenciBenotto | 5010 | 1172 |
| Weeds | 2928 | 354 |
| BIBLESS | 1668 | 33 |
| Baroni | 2770 | 37 |
| Kotlerman | 2940 | 172 |
| Levy | 12602 | 4926 |
| HypeNet-Test | 17670 | 11334 |
| Turney | 2188 | 173 |

Table S12: Dataset sizes. N is the number of word pairs in the dataset, and OOV denotes how many word pairs are not processed.

| | Dataset | | | | | |
|---|---|---|---|---|---|---|
| Method | BLESS | EVALution | LenciBenotto | Weeds | BIBLESS | Baroni |
| Henderson et al. ($D^{\text{Hend.}}$) | 6.4 | 31.6 | 44.8 | 60.8 | 70.5 | **78.3** |
| CMD ($\alpha$=0.15, $s$=1) + $D^{\text{Hend.}}$ | **7.3** | 37.7 | 49.0 | 63.6 | 74.8 | 64.4 |
| CMD ($\alpha$=0.15, $s$=5) + $D^{\text{Hend.}}$ | 6.9 | 39.1 | 49.4 | 64.3 | 74.0 | 65.2 |
| CMD ($\alpha$=0.15, $s$=15) + $D^{\text{Hend.}}$ | 7.0 | 39.8 | 48.5 | 64.7 | 75.0 | 65.6 |
| CMD ($\alpha$=0.5, $s$=1) + $D^{\text{Hend.}}$ | 6.6 | 39.2 | 48.6 | 62.9 | **76.1** | 64.6 |
| CMD ($\alpha$=0.5, $s$=5) + $D^{\text{Hend.}}$ | 5.9 | 40.4 | **49.9** | 65.7 | 73.9 | 67.2 |
| CMD ($\alpha$=0.5, $s$=15) + $D^{\text{Hend.}}$ | 5.5 | **40.5** | 49.5 | **66.2** | 72.8 | 67.4 |

| | Dataset | | | | |
|---|---|---|---|---|---|
| Method | Kotlerman | Levy | Turney | **Avg.Gain** | **Avg. Gain (w/o Baroni)** |
| Henderson et al. ($D^{\text{Hend.}}$) | 34.0 | 11.7 | 56.6 | - | - |
| CMD ($\alpha$=0.15, $s$=1) + $D^{\text{Hend.}}$ | 33.9 | 10.8 | 57.2 | +0.5 | +2.2 |
| CMD ($\alpha$=0.15, $s$=5) + $D^{\text{Hend.}}$ | 34.2 | 11.6 | 57.0 | +0.8 | +2.5 |
| CMD ($\alpha$=0.15, $s$=15) + $D^{\text{Hend.}}$ | 34.9 | 12.3 | **57.3** | +1.2 | **+2.9** |
| CMD ($\alpha$=0.5, $s$=1) + $D^{\text{Hend.}}$ | 34.7 | 10.2 | 56.8 | +0.6 | +2.4 |
| CMD ($\alpha$=0.5, $s$=5) + $D^{\text{Hend.}}$ | 34.6 | 11.3 | 56.5 | +1.2 | +2.7 |
| CMD ($\alpha$=0.5, $s$=15) + $D^{\text{Hend.}}$ | **35.6** | **12.6** | 56.1 | **+1.3** | +2.8 |

Table S13: Comparison of the entailment vectors alone (Hend.) and when used together with our Context Mover's Distance, CMD($\alpha, s$) (where $\alpha$ and $s$ are the PPMI smoothing and shift parameters), in the form of ground cost $D^{\text{Hend.}}$. All of the CMD variants use $K = 100$ clusters. We observe that using our method with the entailment vectors performs better on 8 out of 9 datasets in comparison to just using these vectors alone. Avg. gain refers to the average gain in performance relative to the entailment vectors. Avg. gain w/o Baroni refers to the average performance gain excluding the Baroni dataset. The hyperparameter $\alpha$ refers to the smoothing exponent and $s$ to the shift in the PPMI computation. All scores are AP at all (%).

## S7    QUALITATIVE ANALYSIS OF HYPERNYMY DETECTION

Here, our objective is to qualitatively analyse the particular examples where our method of using Context Mover's Distance (CMD) along with embeddings from Henderson (2017) performs better or worse than just using these entailment embeddings alone.

### S7.1    EVALUATION PROCEDURE

**Comparing by rank.**   Again as in the qualitative analysis with sentence similarity, it doesn't make much sense to compare the raw distance/similarity values between two words as their spread across word pairs can be quite different. We thus compare the ranks assigned to each word pair by both the methods.

**Ground-truth details.**    In contrast to graded ground-truth scores in the previous analysis, here we just have a binary ground truth: 'True' if the hyponym-hypernym relation exists and 'False' when it doesn't. We consider the BIBLESS dataset (Kiela et al., 2015) for this analysis, which has a total of 1668 examples. Out of these, 33 word pairs are not in the vocabulary (see Table S12), so we ignore them for this analysis. Amongst the 1635 examples left, 814 are 'True' and 821 are 'False'. A perfect method should rank the examples labeled as 'True' from 1 to 814 and the 'False' examples from 815 to 1635. Of course, achieving this is quite hard, but the better of the methods should rank as many examples in the desired ranges.

**Example selection criteria.**    We look at the examples where the difference in ranks as per the two methods is the largest. Also, for a few words, we also look at how each method ranks when present as a hypernym and a hyponym. If the difference in ranks is defined as, *CMD rank - Henderson Rank*, we present the top pairs where this difference is most positive and most negative.

### S7.2    RESULTS

For reference on the BIBLESS dataset, CMD performs better than Henderson embeddings quantitatively (cf. Table 2). Let's take a look at some word pairs to get a better understanding.

#### S7.2.1    MAXIMUM POSITIVE DIFFERENCE IN RANKS

These are essentially examples where CMD considers the entailment relation as 'False' while the Henderson embeddings predict it as 'True', and both are most certain about their decisions. Table S14 shows these pairs, along with ranks assigned by the two methods and the ground-truth label for reference.

Some quick observations: many of the word pairs which the Henderson method gets wrong are co-hyponym pairs, such as: ('banjo', 'flute'), ('guitar', 'trumpet'), ('turnip, 'radish'). Additionally, ('bass', 'cello' ), ('creature', 'gorilla'), etc., are examples where the method has to assess not just if the relation exists, but also take into account the directionality between the pair, which the Henderson method seems unable to do.

#### S7.2.2    MAXIMUM NEGATIVE DIFFERENCE IN RANKS

Now the other way around, these are examples where CMD considers the entailment relation as 'True' while the Henderson embeddings predict it as 'False', and both are most certain about their decisions. Table S15 shows these pairs. The examples where CMD performs poorly like, ('box', 'mortality'), ('pistol', 'initiative') seem to be unrelated and we speculate that matching the various contexts or senses of the distributional estimate causes this behavior. One possibility to deal with this can be to take into account the similarity between word pairs in the ground metric. Overall, CMD does a good job at handling these pairs in comparison to the Henderson method.

| Hypernym candidate | Hypernym candidate | Ground Truth | CMD rank | Henderson rank | Better Method |
|---|---|---|---|---|---|
| bass | cello | FALSE | 1346 | 56 | CMD |
| banjo | flute | FALSE | 1312 | 108 | CMD |
| guitar | trumpet | FALSE | 1249 | 52 | CMD |
| trumpet | violin | FALSE | 1351 | 165 | CMD |
| gill | goldfish | FALSE | 1202 | 21 | CMD |
| topside | battleship | FALSE | 1508 | 345 | CMD |
| trumpet | piano | FALSE | 1289 | 126 | CMD |
| washer | dishwasher | FALSE | 1339 | 234 | CMD |
| gun | pistol | FALSE | 1270 | 166 | CMD |
| cauliflower | rainbow | FALSE | 1197 | 136 | CMD |
| hawk | woodpecker | FALSE | 1265 | 210 | CMD |
| garlic | spice | TRUE | 1248 | 204 | Henderson |
| coyote | beast | TRUE | 1096 | 57 | Henderson |
| lizard | beast | TRUE | 1231 | 201 | Henderson |
| turnip | radish | FALSE | 1060 | 39 | CMD |
| creature | gorilla | FALSE | 1558 | 543 | CMD |
| rabbit | squirrel | FALSE | 1260 | 249 | CMD |
| ship | battleship | FALSE | 1577 | 571 | CMD |
| giraffe | beast | TRUE | 1220 | 220 | Henderson |
| coyote | elephant | FALSE | 1017 | 28 | CMD |

Table S14: The top 20 word pairs with **maximum positive difference** in ranks (CMD rank - Henderson rank). The rank is given out of 1635.

| Hyponym candidate | Hypernym candidate | Ground Truth | CMD rank | Henderson rank | Better Method |
|---|---|---|---|---|---|
| box | mortality | FALSE | 116 | 1534 | Henderson |
| radio | device | TRUE | 110 | 1483 | CMD |
| television | system | TRUE | 5 | 1354 | CMD |
| elephant | hospital | FALSE | 52 | 1355 | Henderson |
| pistol | initiative | FALSE | 40 | 1316 | Henderson |
| library | construction | TRUE | 71 | 1335 | CMD |
| radio | system | TRUE | 6 | 1266 | CMD |
| bowl | artifact | TRUE | 223 | 1448 | CMD |
| oven | device | TRUE | 88 | 1279 | CMD |
| bear | creature | TRUE | 324 | 1513 | CMD |
| stove | device | TRUE | 167 | 1356 | CMD |
| saw | tool | TRUE | 461 | 1620 | CMD |
| television | equipment | TRUE | 104 | 1244 | CMD |
| library | site | TRUE | 87 | 1217 | CMD |
| battleship | bus | FALSE | 292 | 1418 | Henderson |
| pistol | device | TRUE | 70 | 1187 | CMD |
| battleship | vehicle | TRUE | 77 | 1175 | CMD |
| bowl | container | TRUE | 333 | 1431 | CMD |
| pub | construction | TRUE | 19 | 1116 | CMD |
| bowl | object | TRUE | 261 | 1334 | CMD |

Table S15: The top 20 word pairs with **maximum negative difference** in ranks (CMD rank - Henderson rank). The rank is given out of 1635.

