# OpenReview forum: "Context Mover's Distance & Barycenters: Optimal transport of contexts for building representations"
_ICLR.cc/2019/Workshop/DeepGenStruct — DeepGenStruct 2019_

### Official Review · AnonReviewer1 · 2019-04-16
**A reasonable proposal for distributionally representing words. Some issues with clarity in writing.**

**Rating:** 3
**Confidence:** 2

**Review:**

Positives:
- The results are straightforward, and compare to the standard baselines (SIF, infersent etc). I think these are not SoTA numbers but the relative comparisons seem fine, especially for a workshop paper. The addition of supplementary ablation experiments is also much appreciated.
- The approach is fairly clear, and there's no real big holes in terms of how the problem is set up and solved.

Conceptual:
- I'm a bit unhappy with how the entailment experiments were set up. The narrative up to the last paragraph ("For this purpose, ..") is fairly clear that one wants to find distributions that are 'contained in the support' of another. You could directly do this on the PPMI matrix you have and define a measure of 'containment' in terms of distributions. Instead of this, you end up plugging in a (heurisitic) ground metric and computing embeddings. This seems haphazard, and I'm not terribly convinced this makes sense. Are you doing consistency checks like making sure you dont have negative-cost-cycles?

Clarity:
- Throughout the paper, there are all sorts of minor unsupported side-remarks and claims that should be stripped. The paper has a fairly straightforward main story and positive results; the addition of these remarks just detract from the rest of the paper. The authors should ask themselves - would I be willing to write a supplemental proof to make this statement precise? or is it just a side-remark I can remove.
- The explanation surrounding equation 4 is fairly confusing. I guess what's happening is that you're taking each context, embedding it to some metric space, and weighting it by its bin count. This is quite the sudden leap, since until this point it wasn't really clear that you were going to embed the contexts with a base embedding. Is this an approximation to the actual thing you want to compute (OT over contexts?) or a way to induce the distances? I think its a little bit of both, but the explanation here needs to be more carefully thought out and laid out. I think it would help to make the inputs and outputs to your method precise.
- 'We also consider adding the information from point estimate into the distributional estimate to get best of both the worlds.' - you should motivate why you want to do this ('best of both worlds' is extremely unclear.. what problem are you solving exactly?). You should then precisely state what you're doing. Your setup Eq (4) is already a bit confusing, so you need to be a bit careful when building on it. I guess the point estimate being added here is the original Glove embedding?
- 'Since the contexts are dense embeddins' - you really need to explain this. A context is a context, not an embedding - I think the more precise statement is that a context can be mapped to a dense embedding. I assume it's something like, you start with point estimates (i.e. glove vectors) of contexts, so you can treat each context as being assigned an associated vector. You then cluster this, and sum over the clusters. I'm also not sure if summing normalized SPPMI values makes sense as an object. Shouldn't you merge the contexts and then compute the SPPMI again over the 'combined' context? Either way, this part can be made much more clear.
- I'm not sure why barycenters is obviously better.. if you have polysemy, you'd want to select the meaning that's implied by all the words, and reject any others. The barycenter does not do this, because you're still incurring costs from the word sense that's not being used in the sentence. I guess it's better than the alternatives?
- In the paragraph connecting SIF and CoMB - i have no idea what the precise connection is. You should write down propositions and equations for any precise statements like this one (or remove it).

Minor comment:
- 'Also, KL-divergence isn’t defined when the supports of distributions under comparison don’t fully overlap' - is false, you only need absolute continuity, meaning you don't need full overlap - just that the support of the distribution inside the log must be a subset of the support of the distribution outside the log.
- 'Hence, one potential application could be in checking for the implicit bias in point estimates (Bolukbasi et al., 2016) and then correcting it via the ground cost.' - this is a pretty vacuous statement - you could have compared pairwise distances for word embeddings to correct it, for example. I honestly think the throwaway comments like this one and the one above should just be stripped from the paper.
- The section 'Relation between the histogram and point estimates.' is similarly vacuous. Yes, count based methods use histograms and neural networks use vectors. Yes, your paper kind of uses both. I really do not think pointing this out adds much insight to your paper. It may be that you had something more profound to say, but it doesn't quite come though.
- 'A practical take-home message of this work thus is to not throw away the co-occurrence information e.g. when using GloVe, but to instead pass it on to our method.' - should be moved to the discussion.
- You may also want to put the timings in the experiment - runtimes are somewhat useless without matching accuracy numbers for approximate algorithms such as sinkhorn. What's the relative (percent) error on your wasserstein distance estimates?
- ' In fact, Figure 3 says it all,' - in fact, figure 3 does not say it all because it's a particular example projected into 2d without very much explanation. In fact i'd argue that it's not a terribly enlightening - what is the ground truth supposed to look like? why is euclidean averaging bad?

---

### Official Review · AnonReviewer2 · 2019-04-16

**Rating:** 4
**Confidence:** 3

**Review:**

This paper proposes to construct word embeddings from a histogram over context words, instead of as point vectors, which allows for measuring distances between two words in terms of optimal transport between the histograms. On sentence similarity and word entailment tasks, the method is competitive with previous approaches, although not by a huge margin.

The paper proposes a method to augment representation of an entity (such as a word) from standard "point in a vector space" to a histogram with bins located at some points in that vector space. In this model, the bins correspond the context objects, the location of which are the standard point embedding of those objects, and the histogram weights correspond to the strength of the contextual association. The distance between two representations is then measured with, Context Mover Distance, based on the theory of optimal transport, which is suitable for computing the discrepancy between distributions.

Pros
- Mathematically elegant method to represent words as distributional estimates of context words.
- Novel idea to use wasserstein barycenter to measure sentence similarity
- Novel idea to use Wasserstein distance for hypernym detection.

Cons:
- Results do not show significant improvement over baselines.
- Potentially complicated for practitioners in the community. Computing CMD and wasserstein barycenters is not trivial and can be inefficient. For this method to be practically useful (and see wide adoption), I believe there has to be a compelling use case for using distributional estimates as oppose to standard point estimates, which isn't demonstrated in the paper. Nevertheless, I believe this paper makes an important contribution.

---

### Decision · Program_Chairs · 2019-04-19
**Acceptance Decision**

Accept